# Neuronal origins of reduced accuracy and biases in economic choices under sequential offers

**Weikang Shi[1], Sebastien Ballesta[1†‡], Camillo Padoa-Schioppa[1,2,3]\***

[1]Department of Neuroscience, Washington University in St. Louis, St. Louis, United States; [2]Department of Economics, Washington University in St. Louis, St. Louis, United States; [3]Department of Biomedical Engineering, Washington University in St. Louis, St. Louis, United States

**\*For correspondence:** camillo@wustl.edu

**Present address:** †Laboratoire de Neurosciences Cognitives et Adaptatives, Strasbourg, France; ‡Centre de Primatologie de l'Université de Strasbourg, Niederhausbergen, France

**Competing interest:** The authors declare that no competing interests exist.

**Abstract** Economic choices are characterized by a variety of biases. Understanding their origins is a long-term goal for neuroeconomics, but progress on this front has been limited. Here, we examined choice biases observed when two goods are offered sequentially. In the experiments, rhesus monkeys chose between different juices offered simultaneously or in sequence. Choices under sequential offers were less accurate (higher variability). They were also biased in favor of the second offer (order bias) and in favor of the preferred juice (preference bias). Analysis of neuronal activity recorded in the orbitofrontal cortex revealed that these phenomena emerged at different computational stages. Lower choice accuracy reflected weaker offer value signals (valuation stage), the order bias emerged during value comparison (decision stage), and the preference bias emerged late in the trial (post-comparison). By neuronal measures, each phenomenon reduced the value obtained on average in each trial and was thus costly to the monkey.

## Editor's evaluation

This manuscript describes three decision biases in value-based choice paradigms. Building on previous work from the lab, the authors focus on neural coding of decision variables in the orbitofrontal cortex of rhesus monkeys, and convincingly argue that different biases arise at different stages of the decision-making process. The reviewers found the study rigorous and believe that the results will be of broad interest. Understanding the neural mechanisms that produce biases in decision-making is an important goal for the field of decision-making and neuroeconomics, and also has relevance to conditions that involve disordered decision-making.

## Introduction

Some of the most mysterious aspects of economic behavior are choice biases documented in behavioral economics (*Camerer et al., 2003*; *Kahneman and Tversky, 2000*; *Lichtenstein and Slovic, 2006*). Standard economic theory fails to account for these effects, and shedding light on their origins is a long-term goal for neuroeconomics (*Camerer et al., 2005*; *Glimcher and Rustichini, 2004*). Progress on this front has been relatively modest, largely because the neural underpinnings of (even simple) choices were poorly understood until recently. However, the last 15 years have witnessed substantial advances. An important turning point was the development of experimental protocols in which subjects choose between different goods and relative subjective values are inferred from choices. Decision variables defined from these values are used to interpret neural activity (*Kable and Glimcher, 2007*; *Padoa-Schioppa and Assad, 2006*; *Plassmann et al., 2007*). Studies that adopted

this paradigm showed that neurons in numerous brain regions represent the values of offered and chosen goods (*Amemori and Graybiel, 2012*; *Cai et al., 2011*; *Cai and Padoa-Schioppa, 2012*; *Hosokawa et al., 2013*; *Jezzini and Padoa-Schioppa, 2020*; *Kim et al., 2008*; *Lak et al., 2014*; *Levy et al., 2010*; *Louie and Glimcher, 2010*; *Padoa-Schioppa and Assad, 2006*; *Pastor-Bernier et al., 2019*; *Shenhav and Greene, 2010*). Furthermore, recent experiments using electrical stimulation showed that offer values encoded in the orbitofrontal cortex (OFC) are causally linked to choices (*Ballesta et al., 2020*). These results are of high significance for three reasons.

First, the identification in OFC and other brain regions of distinct groups of neurons encoding different decision variables is essential to ultimately understand the neural circuit and the mechanisms through which economic decisions are formed.

Second, in a more conceptual sense, the results summarized above provide a long-sought validation for the construct of value. The proposal that choices entail computing and comparing subjective values was put forth by early economists such as Bernoulli and Bentham (*Niehans, 1990*). Although this idea has remained influential, values defined at the behavioral level suffer from a fundamental problem of circularity. On the one hand, choices supposedly maximize values; on the other hand, values cannot be measured behaviorally independent of choices (*Samuelson, 1938*). Because of this problem, the construct of value gradually lost centrality in economic theory. Thus, in the standard neoclassic formulation choices are 'as if' driven by values, but there is no commitment to the idea that agents actually compute values (*Samuelson, 1947*). In this perspective, the fact that neuronal firing rates in any brain region are linearly related to values defined at the behavioral level constitutes powerful evidence that choices indeed entail the computation of values (*Camerer, 2008*).

Third and less frequently discussed, the identification of neurons encoding offer values and other decision variables, together with some rudimentary understanding of the decision circuit, provides the opportunity to break the circularity problem described above. To appreciate this point, consider the fact that economic choices are often affected by seemingly idiosyncratic biases. For example, while choosing between two options offered sequentially, people and monkeys typically show a bias favoring the second option (*Ballesta and Padoa-Schioppa, 2019*; *Krajbich et al., 2010*; *Rustichini et al., 2021*). This order bias might occur for at least two reasons. (1) Subjects might assign a higher value to any given good if that good is offered second. (2) Alternatively, subjects might assign identical values independent of the presentation order, and the bias might emerge downstream of valuation, for example during value comparison. In the latter scenario, by introducing the order bias, the decision process would actually fail to maximize the value obtained by the agent. Due to the circularity problem described above, these two hypotheses are ultimately not distinguishable based on behavior alone. However, access to a credible neural measure for the offer values makes it possible, at least in principle, to disambiguate between them. The results presented in this study build on this fundamental idea.

We focused on choice biases measured when two goods are offered sequentially. In the experiments, monkeys chose between two juices offered in variable amounts. In each session, we randomly interleaved two types of trials referred to as two tasks. In Task 1, offers were presented simultaneously; in Task 2, offers were presented in sequence. Comparing choices across tasks revealed three phenomena. (1) Monkeys were substantially less accurate (higher choice variability) in Task 2 (sequential offers) compared to Task 1 (simultaneous offers). (2) Choices in Task 2 were biased in favor of the second offer (order bias). (3) Choices in Task 2 were biased in favor of the preferred juice (preference bias). These effects are especially interesting because in most daily situations offers available for choice appear or are examined sequentially. Thus, we investigated the neuronal origins of these phenomena.

Neuronal recordings focused on the OFC. Earlier work on choices under simultaneous offers identified in this area different groups of cells encoding individual offer values, the binary choice outcome (chosen juice), and the chosen value (*Padoa-Schioppa, 2013*; *Padoa-Schioppa and Assad, 2006*). Furthermore, previous analyses indicated that choices under sequential offers engage the same neuronal populations (*Ballesta and Padoa-Schioppa, 2019*; *Shi et al., 2022a*). In other words, the cell groups labeled *offer value*, *chosen juice* and *chosen value* can be identified in either choice task and appear to preserve their functional role. In first approximation, the variables encoded in OFC capture both the input (offer values) and the output (chosen juice, chosen value) of the choice process, suggesting that the cell groups identified in this area constitute the building blocks of a decision circuit (*Padoa-Schioppa and Conen, 2017*). A series of experimental (*Ballesta et al., 2021*; *Camille et al.,*

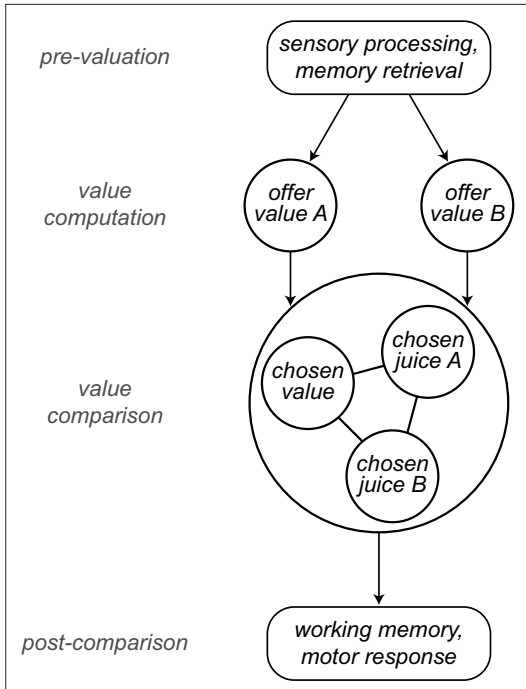

**Figure 1.** Computational framework. Information about sensory input, stored memory, and the motivational state is integrated during the computation of offer values. In orbitofrontal cortex (OFC), *offer value* cells provide the primary input to a decision circuit composed of *chosen juice* cells and *chosen value* cells. The detailed structure of the decision circuit is not well understood, but previous work indicates that decisions under sequential offers rely on circuit inhibition. In essence, neurons encoding the value of the first offer (offer1) indirectly impose a negative offset on the activity of *chosen juice* cells associated with the second offer (offer2). Notably, this circuit might also subserve working memory. The decision output, captured by the activity of *chosen juice* cells, informs other brain regions that transform it into a suitable action plan. Choice measured behaviorally is ultimately defined by the motor response. This framework highlights the fact that choice biases and/or noise might emerge at multiple computational stages. The arrows indicated here capture only the primary connections.

2011; *Rich and Wallis, 2016*) and theoretical (*Friedrich and Lengyel, 2016*; *Rustichini and Padoa-Schioppa, 2015*; *Solway and Botvinick, 2012*; *Song et al., 2017*; *Zhang et al., 2018*) results support this view. Here, we put forth a more articulated computational framework. In our account, different groups of OFC neurons participate in value computation and value comparison, and these processes are embedded in an ensemble of mental operations taking place before, during, and after the decision itself. In this view, sensory information, memory traces, and internal states are processed upstream of OFC and integrated in the activity of *offer value* cells. These neurons provide the primary input to a circuit formed by *chosen juice* cells and *chosen value* cells, where values are compared. The output of this circuit feeds brain regions involved in working memory and the construction of action plans (*Figure 1*).

This framework guided a series of analyses relating the activity of each cell group to the choice biases described above. Our results revealed that different phenomena emerged at different computational stages. The lower choice accuracy observed under sequential offers reflected weaker offer value signals (valuation stage). Conversely, the order bias did not have neural correlates at the valuation stage, but rather emerged during value comparison (decision stage). Finally, the preference bias did not have neural correlates at the valuation stage or during value comparison; it emerged late in the trial, shortly before the motor response.

## Results
### Reduced accuracy and biases in choices under sequential offers
Two monkeys participated in the experiments. In each session, they chose between two juices labeled A and B, with A preferred. Offers were represented by sets of colored squares on a monitor, and animals indicated their choice with a saccade. In each session, two choice tasks were randomly interleaved. In Task 1, offers were presented simultaneously (*Figure 2A*); in Task 2, offers were presented in sequence (*Figure 2B*). A cue displayed at the beginning of the trial revealed to the animal the task for that trial. Offers varied from trial to trial, and we indicate the quantities offered in any given trial with $q_A$ and $q_B$. An 'offer type' was defined by two quantities [$q_A$, $q_B$], and the same offer types were used for the two tasks in each session. For Task 2, trials in which juice A was offered first and trials in which juice B was offered first are referred to as 'AB trials' and 'BA trials', respectively. The first and second offers are referred to as 'offer1' and 'offer2', respectively.

The data set included 241 sessions (101 from monkey J, 140 from monkey G; see Methods). Sessions lasted for 217–880 trials (mean ± std = 589 ± 160). For each session, we analyzed choices in the two tasks separately using probit regressions. For Task 1 (simultaneous offers), we used the following model:

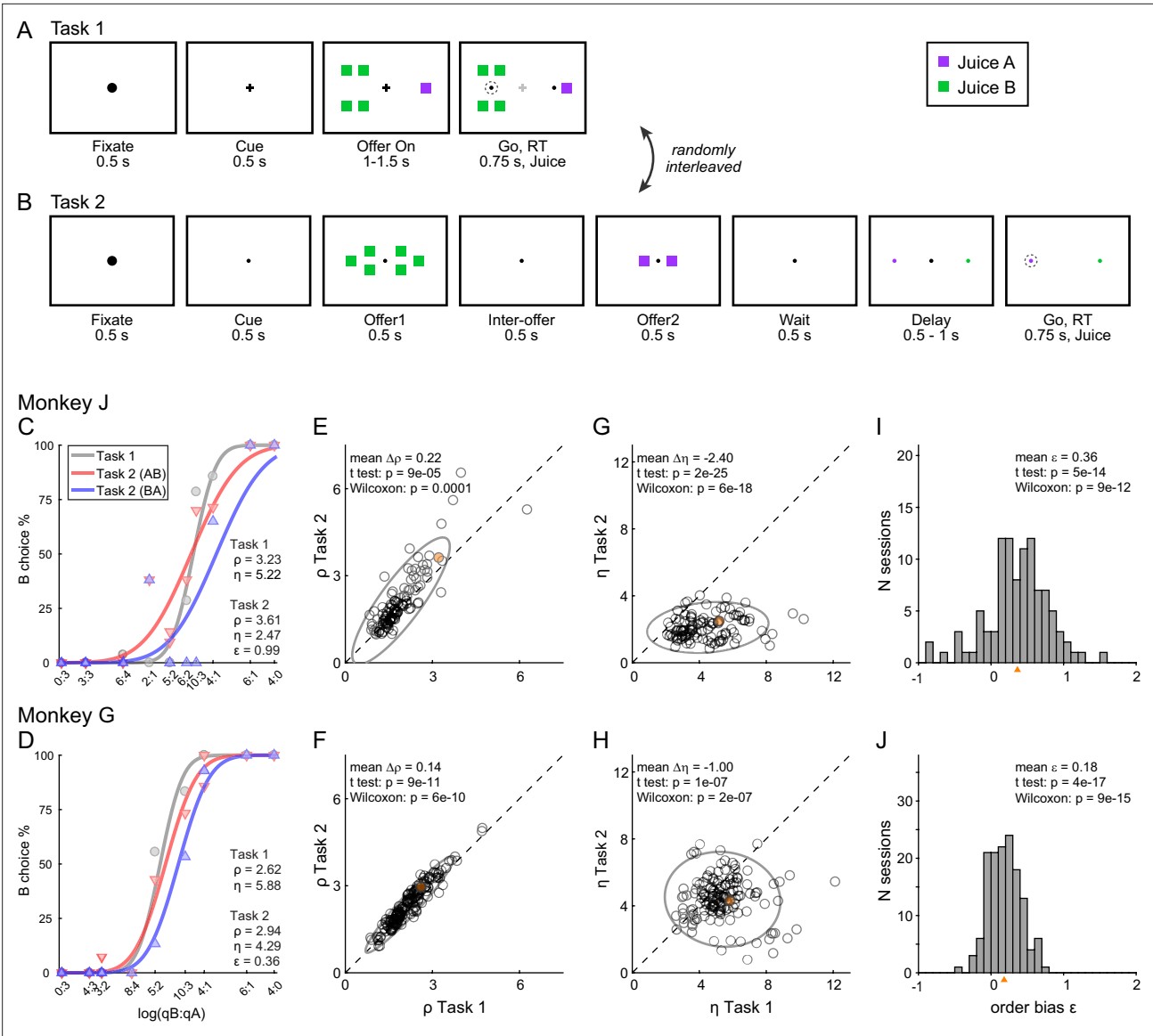

**Figure 2.** Experimental design and choice biases. (**A, B**) Experimental design. Animals chose between two juices offered in variable amounts. Offers were represented by sets of color squares. For each offer, the color indicated the juice type and the number of squares indicated the juice amount. In each session, trials with Tasks 1 and 2 were randomly interleaved. In Task 1, two offers appeared simultaneously on the left and right sides of the fixation point. In Task 2, offers were presented sequentially, spaced by an interoffer delay. After a wait period, two saccade targets matching the colors of the offers appeared on the two sides of the fixation point. The left/right configuration in Task 1, the presentation order in Task 2, and the left/right position of the saccade targets in Task 2 varied randomly from trial to trial. In any session, the same set of offer types was used for both tasks. (**C**) Example session 1. The percent of B choices (y-axis) is plotted against the log quantity ratio (x-axis). Each data point indicates one offer type in Task 1 (gray circles) or Task 2 (red and blue triangles for AB trials and BA trials, respectively). Sigmoids were obtained from probit regressions. The relative value ($\rho$) and sigmoid steepness ($\eta$) measured in each task and the order bias ($\varepsilon$) measured in Task 2 are indicated. In this session, the animal presented all three biases. Compared to Task 1, choices in Task 2 were less accurate ($\eta_{Task2} < \eta_{Task1}$) and biased in favor of juice A ($\rho_{Task2} > \rho_{Task1}$; preference bias). Furthermore, choices in Task 2 were biased in favor of offer2 ($\varepsilon > 0$; order bias). (**D**) Example session 2. Same format as panel C. (**E, F**) Comparing relative value across choice tasks. Each data point represents one session and gray ellipses indicate 90% confidence intervals. For both monkeys, relative values in Task 2 (y-axis) were significantly higher than in Task 1 (x-axis). Furthermore, the main axis of each ellipse was rotated counterclockwise compared to the identity line. (**G, H**) Comparing the sigmoid steepness across choice tasks. For both monkeys, sigmoids were consistently shallower (smaller $\eta$) in Task 2 compared to Task 1. (**I, J**) Order bias, distribution across sessions. Both monkeys presented a consistent bias favoring offer2 (mean($\varepsilon$) > 0). Panels **C, E, G, and I** are from monkey J (N = 101 sessions); panels **D, F, H, and J** are from monkey G (N = 140 sessions). Sessions shown in panels **C and D** are highlighted in yellow in panels **E, F, G, and H**. Triangles in panels **I and J** indicate the mean. Statistical tests and exact p values are indicated in each panel.

$$Choice\ B = \Phi(X)$$
$$X = a_0 + a_1\ log(q_B/q_A)$$

(1)

where *Choice B* = 1 if the animal chose juice B and 0 otherwise, $\Phi$ was the cumulative function of the standard normal distribution, and $q_A$ and $q_B$ were the quantities of juices offered on any given trial. From the fitted parameters $a_0$ and $a_1$, we derived measures for the relative value of the two juices $\rho_{Task1}$ = exp($-a_0/a_1$) and for the sigmoid steepness $\eta_{Task1} = a_1$. Intuitively, the relative value was the quantity ratio $q_B/q_A$ that made the animal indifferent between the two juices, and the sigmoid steepness was inversely related to choice variability.

For Task 2 (sequential offers), we used the following model:

$$Choice\ B = \Phi(X)$$
$$X = a_2 + a_3\ log(q_B/q_A) + a_4\ (\delta_{order,AB} - \delta_{order,BA})$$

(2)

where $\delta_{order,AB}$ = 1 for AB trials and 0 otherwise, and $\delta_{order,BA} = 1 - \delta_{order,AB}$. In essence, AB trials and BA trials were analyzed separately but assuming that the two sigmoids were parallel. From the fitted parameters $a_2$, $a_3$, and $a_4$, we derived measures for the relative value of the two juices $\rho_{Task2}$ = exp($-a_2/a_3$), for the sigmoid steepness $\eta_{Task2} = a_3$, and for the order bias $\varepsilon = 2\ \rho_{Task2}\ a_4/a_3$. Intuitively, the order bias was a bias favoring the first or the second offer. Specifically, $\varepsilon < 0$ indicated a bias favoring offer1; $\varepsilon > 0$ indicated a bias favoring offer2. We also defined relative values specific to AB trials and BA trials as $\rho_{AB}$ = exp($-(a_2 + a_4)/a_3$) and $\rho_{BA}$ = exp($-(a_2 - a_4)/a_3$). Of note, the order bias was defined such that

$$\epsilon \approx \rho_{BA} - \rho_{AB}$$

(3)

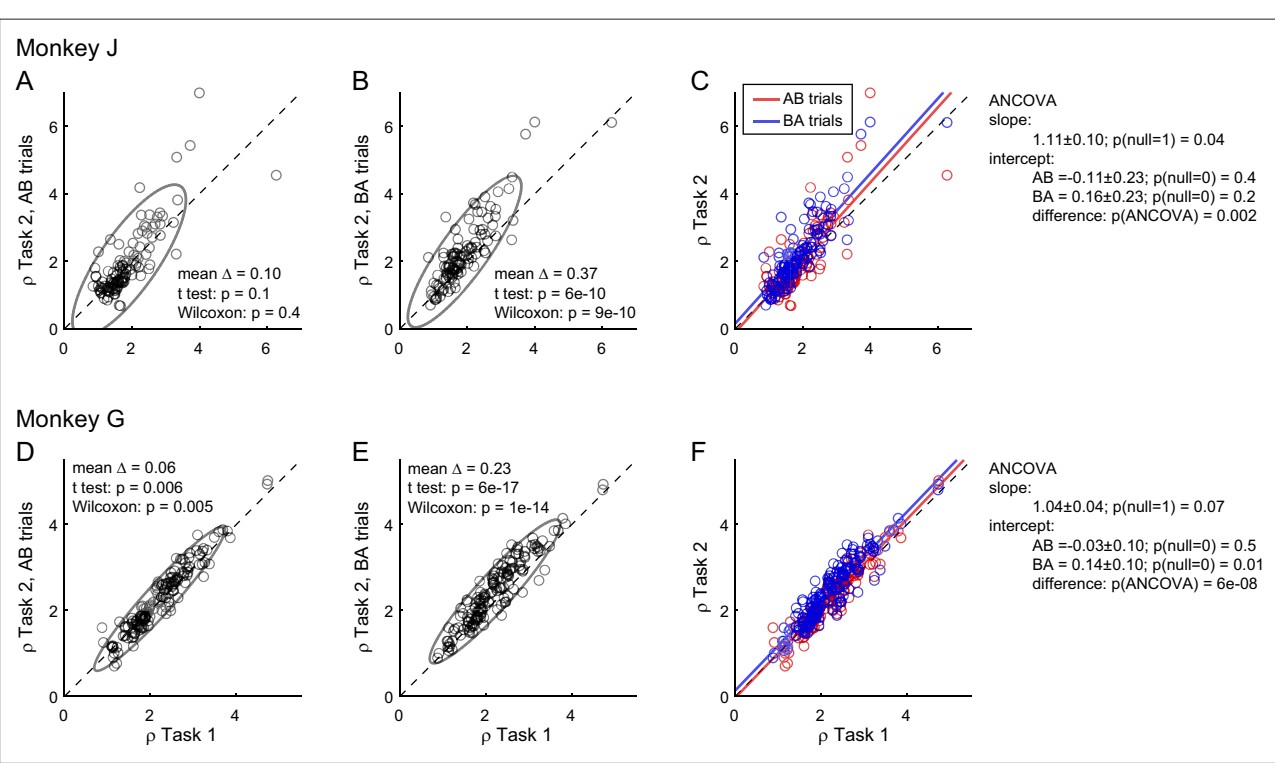

**Figure 3.** Order bias and preference bias. (**A–C**) Monkey J (*N* = 101 sessions). In panels A and B, $\rho_{Task2,AB}$ and $\rho_{Task2,BA}$ (*y*-axis) are plotted against $\rho_{Task1}$ (*x*-axis). Each data point represents one session and gray ellipses indicate 90% confidence intervals. The main axis of both ellipses is rotated counterclockwise compared to the identity line (preference bias). In addition, the ellipse in panel B is displaced upwards compared to that in panel A (order bias). In panel C, the same data are pooled and color coded. The two lines are from an ANCOVA (covariate: order; parallel lines). The regression slope is significantly >1 (preference bias) and the two intercepts differ significantly from each other (order bias). (**D–F**) Monkey G (*N* = 140 sessions). Same format. The results closely resemble those for monkey J but the preference bias is weaker.

The experimental design gave us the opportunity to compare choices across tasks independently of factors such as selective satiation or changes in the internal state. The relative values measured in the two tasks were highly correlated (*Figure 2E, F*). At the same time, our analyses revealed three interesting phenomena. First, for both animals, sigmoids measured in Task 2 were significantly shallower compared to Task 1 (*Figure 2G, H*). In other words, presenting offers in sequence reduced choice accuracy. Second, in Task 2, both animals showed a consistent order bias favoring offer2 (*Figure 2I, J*). Third, in both animals, relative values in Task 2 were significantly higher than in Task 1 ($\rho_{Task2} > \rho_{Task1}$), and this effect increased with the relative value (*Figure 2E, F*). In other words, the ellipse marking the 90% confidence interval for the joint distribution of relative values laid above the identity line and was rotated counterclockwise compared to the identity line.

To further investigate the differences in relative values measured across tasks, we quantified them separately in AB trials and BA trials in each monkey. We thus examined the relation between $\rho_{Task1}$ and $\rho_{Task2,AB}$ and, separately, that between $\rho_{Task1}$ and $\rho_{Task2,BA}$ (*Figure 3*). In both animals and in both sets of trials, the ellipse marking the 90% confidence interval was rotated counterclockwise compared to the identity line. Furthermore, the ellipse measured for BA trials was higher than that for AB trials. We quantified these observations with an analysis of covariance (ANCOVA) using the presentation order (AB and BA) as a covariate and imposing parallel lines (*Figure 3C, F*). In both animals, the two regression lines were significantly distinct (difference in intercept >0, $p \leq 0.002$ in each animal). This result confirmed the presence of an order bias favoring offer2 in Task 2. Concurrently, in both animals the regression slope was significantly >1 ($p \leq 0.04$ in each animal; ellipse rotation). This result indicated that the animals had an additional bias favoring juice A in Task 2, and that this bias increased as a function of the relative value $\rho$. We refer to this phenomenon as the preference bias.

## Computational framework

The following sections present a series of results on the neuronal origins of these behavioral phenomena. We begin by discussing the computational framework for the analyses.

Economic choice is thought to entail two stages: values are assigned to the available offers and a decision is made by comparing values. Importantly, in our tasks and in most circumstances, choices elicit an ensemble of mental operations taking place before, during, and after the computation and comparison of offer values. Upstream of valuation, choices examined here entail the sensory processing of visual stimuli and the retrieval from memory of relevant information (e.g., the association between color and juice type). Downstream of value comparison, the decision outcome must guide a suitable motor response. In addition, performance in Task 2 requires holding in working memory the value of offer1 until offer2, remembering the decision outcome for an additional delay, and mapping that outcome onto the appropriate saccade target (*Figure 2B*). In principle, choice biases could emerge at any of these computational stages. Likewise, each of these mental operations could be noisy and thus contribute to choice variability.

Neuronal activity in OFC does not capture all of these processes. However, previous work indicates that neurons in this area participate both in value computation and value comparison. In the framework proposed here (*Figure 1*), sensory and limbic areas feed *offer value* cells, where values are integrated. In turn, *offer value* cells provide the primary input to a neural circuit constituted by *chosen juice* cells and *chosen value* cells, where decisions are formed. Finally, the decision circuit is connected with downstream areas, such as lateral prefrontal cortex, engaged in working memory and in transforming choice outcomes into suitable action plans. This scheme reflects the anatomical connectivity of OFC and other prefrontal regions (*Carmichael and Price, 1995a*; *Carmichael and Price, 1995b*; *Petrides and Pandya, 2006*; *Saleem et al., 2014*; *Takahara et al., 2012*); it is motivated by neurophysiology results from OFC (*Ballesta et al., 2020*; *Rich and Wallis, 2016*) and connected areas (*Cai and Padoa-Schioppa, 2014*; *Sasikumar et al., 2018*); and it is consistent with computational models of economic decisions (*Friedrich and Lengyel, 2016*; *Rustichini and Padoa-Schioppa, 2015*; *Solway and Botvinick, 2012*; *Song et al., 2017*; *Yim et al., 2019*; *Zhang et al., 2018*).

Of note, both *offer value* and *chosen value* cells encode subjective values. However, in the framework of *Figure 1*, *offer value* cells express a pre-decision value, while *chosen value* cells express a value emerging during the decision process. Conversely, the activity of *chosen juice* cells captures the evolving commitment to a particular choice outcome. In this framework, suitable analyses of neuronal

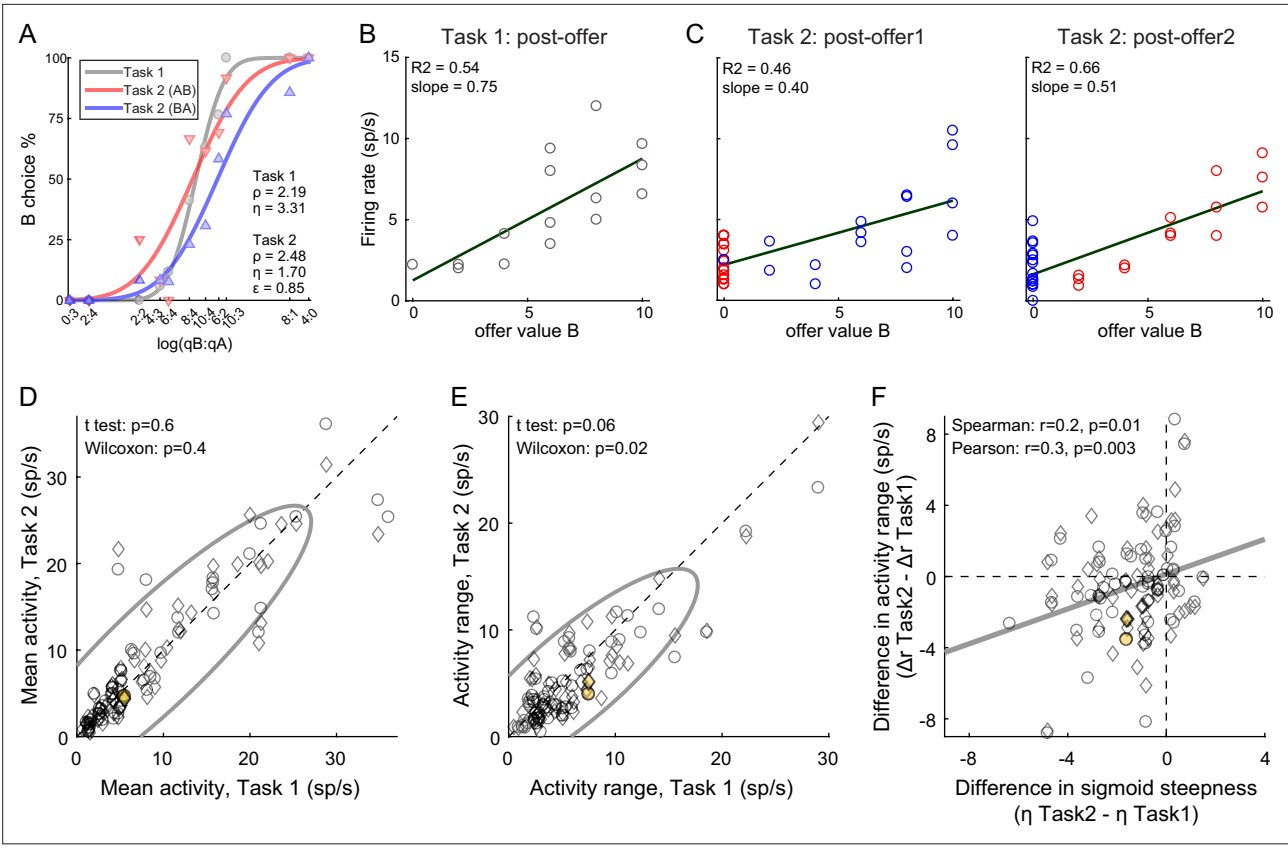

**Figure 4.** Lower choice accuracy in Task 2 reflects weaker offer value signals. (**A–C**) Weaker offer value signals in Task 2, example cell. Panel A illustrates the choice pattern. Panel B illustrates the neuronal response measured in Task 1 (post-offer time window). Each data point represents one trial type. In C, two panels illustrate the neuronal responses measured in Task 2 (post-offer1 and post-offer2 time windows). Each data point represents one trial type; red and blue colors are for AB and BA trials, respectively. In panels B and C, firing rates (*y*-axis) are plotted against variable *offer value B* and gray lines are from linear regressions. Notably, the cell has lower activity range in Task 2 than in Task 1. (**D, E**) Weaker offer value signals in Task 2, population analysis (*N* = 109 *offer value* cells). The two panels illustrate the results for the mean activity and the activity range, respectively. In each panel, *x*- and *y*-axis represent measures obtained in Task 1 and Task 2, respectively. Each data point represents one cell. For each cell, we examined one time window (post-offer) in Task 1 and two time windows (post-offer1 and post-offer2) in Task 2. Circles and diamonds refer to post-offer1 and post-offer2 time windows, respectively. Measures of mean activity measured in the two tasks (panel D) were statistically indistinguishable. In contrast, activity ranges (panel E) were significantly reduced in Task 2 compared to Task 1. Statistical tests and exact p values are indicated in each panel. The example cell shown in panels A–C is highlighted in orange in panels D and E. (**F**) Offer value signals and choice accuracy (*N* = 109 cells). For each *offer value* cell, we computed the activity range Δr in each task (see Methods). Here, the difference in activity range Δr = Δr_{Task2} − Δr_{Task1} (*y*-axis) is plotted against the difference in sigmoid steepness Δη = η_{Task2} − η_{Task1} measured in the same session (*x*-axis). The two measures were significantly correlated across the population. The gray line in panel F is from a linear regression. This analysis was restricted to 53 cells significantly tuned in the post-offer time window (Task 1) and post-offer1 time window (Task 2), and 56 cells significantly tuned in the post-offer time window (Task 1) and post-offer2 time window (Task 2).

The online version of this article includes the following figure supplement(s) for figure 4:

**Figure supplement 1.** Comparing tuning functions across choice tasks.

activity may reveal whether particular choice biases emerge during valuation, during value comparison, or in subsequent computational stages.

## Reduced accuracy under sequential offers emerged at the valuation stage

Other things equal, choices under sequential offers (Task 2) were significantly less accurate than choices under simultaneous offers (Task 1; *Figure 2*). We first investigated the neural origins of this phenomenon.

The primary data set examined in this study included 183 *offer value* cells, 160 *chosen juice* cells, and 174 *chosen value* cells (see Methods). Comparing neuronal responses across tasks, we noted that offer value signals in Task 2 were significantly weaker than in Task 1. *Figure 4A, C* illustrates one

example cell. In both tasks, this neuron encoded the *offer value B*. However, the activity range (see Methods) measured in Task 2 was smaller than that measured in Task 1. This effect was also observed at the population level. For this analysis, we pooled *offer value* cells associated with juices A and B, and with positive or negative encoding (see Methods). For Task 1, we focused on the post-offer time window; for Task 2, we focused on post-offer1 and post-offer2 time windows, pooling trial types from both windows. For each cell, we imposed that the response be significantly tuned in these time windows in each task, and we quantified the mean activity and the activity range ($\Delta r$, see Methods). At the population level, the mean activity did not differ significantly across tasks (p = 0.6, *t*-test; p = 0.4, Wilcoxon test, *Figure 4D*). In contrast, the activity range was significantly lower in Task 2 compared to Task 1 ($\Delta r_{Task2} < \Delta r_{Task1}$; p = 0.06, *t*-test; p = 0.02, Wilcoxon test *Figure 4E*). In other words, offer value signals were weaker in Task 2 compared to Task 1.

The activity of *offer value* cells is causally related to choices (*Ballesta et al., 2020*). Furthermore, for given value range and mean activity, the activity range determines the neuronal signal-to-noise ratio. Indeed, we previously found that decreases in the encoding slope of *offer value* cells due to range adaptation reduce choice accuracy (*Conen and Padoa-Schioppa, 2019*; *Rustichini et al., 2017*). Along similar lines, we inquired whether the difference in choice accuracy measured across tasks (*Figure 2G, H*) might be explained, at last partly, by differences in neuronal activity range (*Figure 4E*). We thus examined the relation between the difference in sigmoid steepness ($\Delta \eta = \eta_{Task2} - \eta_{Task1}$) and the difference in activity range ($\Delta r = \Delta r_{Task2} - \Delta r_{Task1}$). The two measures were positively correlated (Spearman r = 0.2, p = 0.01; Pearson r = 0.3, p = 0.003; *Figure 4F*). In other words, the drop in choice accuracy observed in Task 2 compared to Task 1 correlated with weaker offer value signals. A similar analysis of *chosen value* cells found that the activity range $\Delta r$ was reduced in Task 2 compared to Task 1. However, this effect and the drop in choice accuracy were not significantly correlated (*Figure 4—figure supplement 1*).

In conclusion, the lower choice accuracy measured in Task 2 compared to Task 1 correlated with weaker offer value signals in OFC. Thus, this behavioral phenomenon emerged, at least partly, during valuation.

**Table 1.** Neuronal encoding of decision variables in the two choice tasks.

The table summarizes the results of a previous report (*Shi et al., 2022a*). Under simultaneous offers, different groups of orbitofrontal cortex (OFC) neurons encode different decision variables, each with positive or negative sign (indicated here with + and −). In first approximation, each cell encodes the same variable across time windows. Under sequential offers, OFC neurons encode different variables in different time windows. However, the vast majority of them present one of eight specific patterns of variables, referred to as variable 'sequences' and detailed here. Furthermore, there is a clear correspondence between neurons encoding a particular variable in Task 1 and neurons encoding a particular sequence in Task 2. Hence, we can refer to different cell groups in OFC using the standard nomenclature originally defined for Task 1.

| Task 1 | Task 2 | | |
| --- | --- | --- | --- |
| | **Post-offer1** | **Post-offer2** | **Post-juice** |
| *offer value A +* | *offer value A | AB +* | *offer value A | BA +* | *chosen value A +* |
| *offer value A −* | *offer value A | AB −* | *offer value A | BA −* | *chosen value A −* |
| *offer value B +* | *offer value B | BA +* | *offer value B | AB +* | *chosen value B +* |
| *offer value B −* | *offer value B | BA -* | *offer value B | AB −* | *chosen value B −* |
| *chosen juice A* | *AB | BA +* | *AB | BA −* | *chosen juice A* |
| *chosen juice B* | *AB | BA −* | *AB | BA +* | *chosen juice B* |
| *chosen value +* | *offer value1 +* | *offer value2 +* | *chosen value +* |
| *chosen value −* | *offer value1 −* | *offer value2 −* | *chosen value −* |

## The order bias emerged during value comparison

The next series of analyses focused on the neural origins of the order bias ($\varepsilon$). Since this phenomenon pertains only to choices under sequential offers, we included in the analyses an additional data set recorded in the same animals performing only Task 2 (see Methods).

In the framework of *Figure 1*, we first inquired whether the order bias emerged during valuation. If this was the case, for any given good, *offer value* cells should encode a higher value when the good is presented as offer2. To test this hypothesis, we pooled *offer value* cells associated with the two juices. For each neuron, 'E' indicated the juice encoded by the cell and 'O' indicated the other juice. We thus refer to EO trials and OE trials. For any given cell, we compared the response recorded in EO trials (post-offer1 time window) with the response recorded in OE trials (post-offer2 time window). If the order bias emerged during valuation, the mean activity and/or the activity range should be higher for the latter (*Figure 5—figure supplement 1A*). Contrary to this prediction, across a population of 128 cells, we did not find any systematic difference in mean activity or activity range (*Figure 5—figure supplement 1B, C*). Furthermore, the difference between the activity parameters measured in OE and EO trials did not correlate with the order bias (*Figure 5—figure supplement 1D*). In conclusion, assigned values did not depend on the presentation order.

We next examined whether the order bias emerged during value comparison. If so, the bias should be reflected in the activity of both *chosen juice* and *chosen value* cells (*Figure 1*). For *chosen value* cells, the hypothesis might be tested noting that in post-offer1 and post-offer2 time windows these neurons encoded the value currently offered independently of the juice type (*Table 1*). Thus, the activity measured in these time windows in AB and BA trials provided neuronal measures for the relative values of the two juices. More specifically, for each *chosen value* cell, we derived the two measures $\rho^{neuronal}_{AB}$ and $\rho^{neuronal}_{BA}$ for AB trials and BA trials, respectively (*Figure 5A*; see Methods). We also defined the difference $\Delta\rho^{neuronal} = \rho^{neuronal}_{BA} - \rho^{neuronal}_{AB}$. We recall that the order bias ($\varepsilon$) was essentially equal to the difference between the relative values measured behaviorally in BA and AB trials (*Equation 3*). Thus, assessing whether the activity of *chosen value* cells reflected the order bias amounts to testing the relation between $\Delta\rho^{neuronal}$ and $\varepsilon$.

We conducted a population analysis of 96 *chosen value* cells. Confirming previous results (*Padoa-Schioppa and Assad, 2006*), neuronal and behavioral measures of relative value were highly correlated. Similarly, the two neuronal measures of relative value, $\rho^{neuronal}_{AB}$ and $\rho^{neuronal}_{BA}$, were correlated with each other (*Figure 5B*). Most importantly, the difference $\Delta\rho^{neuronal}$ and the order bias $\varepsilon$ were significantly correlated across the population (Spearman $r = 0.3$, $p = 0.007$; Pearson $r = 0.2$, $p = 0.02$; *Figure 5C*). Hence, session-to-session fluctuations in the activity of *chosen value* cells correlated with fluctuations in the order bias.

Further insights on the order bias came from the analysis of *chosen juice* cells. Again, for each neuron, E and O indicated the juice encoded by the cell and the other juice, respectively. A previous study found that the baseline activity of *chosen juice* cells recorded in OE trials immediately before offer2 was negatively correlated with the value of offer1 (i.e., the value of the other juice) – a phenomenon termed circuit inhibition (*Ballesta and Padoa-Schioppa, 2019*). If the decision is conceptualized as the evolution of a dynamic system (*Rustichini and Padoa-Schioppa, 2015*; *Wang, 2002*), circuit inhibition sets the system's initial conditions and is thus integral to value comparison. In this account, the evolving decision is essentially captured by the activity of *chosen juice* cells in OE trials, which reflects a competition between the negative offset set by the value of offer1 (initial condition) and the incoming signal encoding the value of offer2. If so, the intensity of circuit inhibition should be negatively correlated with the order bias.

We tested this prediction as follows. First, we replicated previous findings and confirmed the presence of circuit inhibition in our primary data set (*Figure 6A*). We then focused on a 300-ms time window starting 250 ms before offer2 onset. For each *chosen juice* cell, we regressed the firing rate against the normalized offer1 value (see Methods). Thus, the regression slope $c_1$ quantified circuit inhibition for individual cells. Across a population of 295 *chosen juice* cells, mean($c_1$) was significantly $<0$ ($p = 5 \times 10^{-6}$, *t*-test; $p = 9 \times 10^{-8}$, Wilcoxon test; *Figure 6B*). Third, we examined the relation between circuit inhibition ($c_1$) and the order bias ($\varepsilon$). Confirming the prediction, the two measures were significantly correlated across the population (Spearman $r = 0.1$, $p = 0.02$; Pearson $r = 0.1$, $p = 0.02$; *Figure 6C*). In other words, stronger circuit inhibition (more negative $c_1$) corresponded to a weaker order bias (smaller $\varepsilon$).

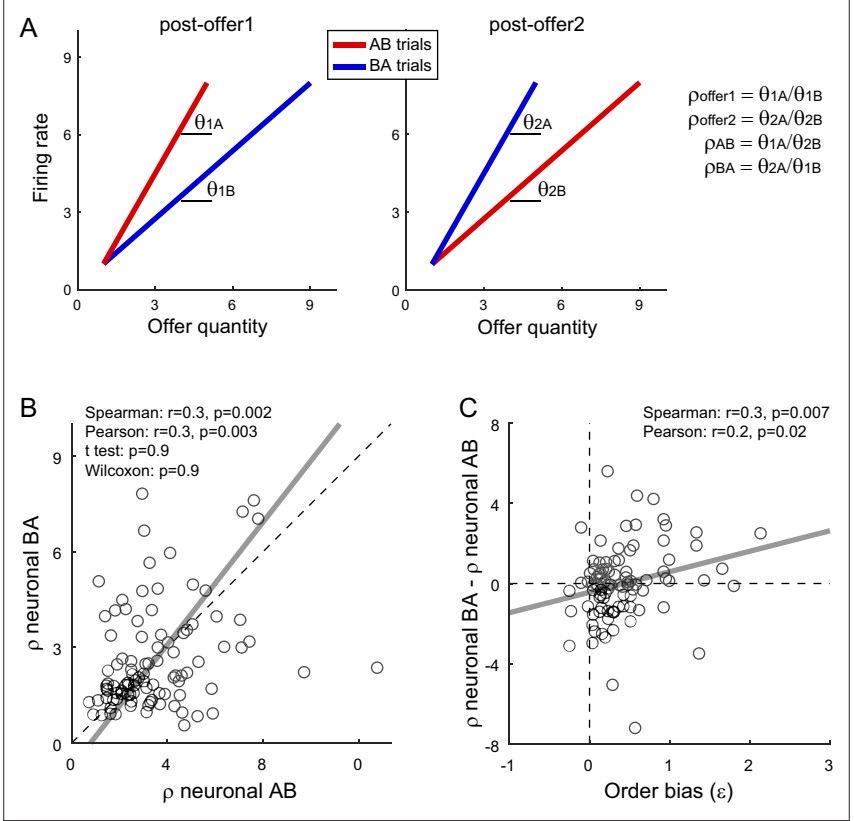

**Figure 5.** Fluctuations in order bias and fluctuations in the activity of *chosen value* cells. (**A**) Neuronal measures of relative value. The two panels represent in cartoon format the response of a *chosen value* cell in the post-offer1 and post-offer2 time window (Task 2). In each of these time windows, *chosen value* cells encode the value of the offer on display. Here, the two axes correspond to the firing rate (*y*-axis) and to the offered juice quantity (*x*-axis). The two colors correspond to the two orders (AB, BA). In each time window, two linear regressions provide two slopes, proportional to the value of the two juices. From the four measures $\theta_{1A}$ (left panel, red), $\theta_{1B}$ (left panel, blue), $\theta_{2A}$ (right panel, blue), and $\theta_{2B}$ (right panel, red), we derive four neuronal measures of relative value (Methods, *Equations 13–16*). (**B**) Neuronal measures of relative value in AB trials and BA trials (*N* = 96 cells). The *x*- and *y*-axis correspond to $\rho^{neuronal}_{AB}$ and $\rho^{neuronal}_{BA}$, respectively. Each data point represents one cell. The two measures are strongly correlated. The gray line is from a Deming regression. (**C**) Fluctuations of relative value and fluctuations in order bias (*N* = 96 cells). For each *chosen value* cell, we quantified the difference in the neuronal measure of relative value $\Delta\rho^{neuronal} = \rho^{neuronal}_{AB} - \rho^{neuronal}_{BA}$. Here, the *x*-axis is the order bias ($\varepsilon$), the *y*-axis is $\Delta\rho^{neuronal}$, and each data point corresponds to one cell. The gray line is from a linear regression. Statistical tests and exact p values are indicated in each panel. This analysis was restricted to 96 cells that had significant $\theta_{1A}$, $\theta_{1B}$, $\theta_{2A}$, and $\theta_{2B}$. Fluctuations of $\Delta\rho^{neuronal}$ correlated with fluctuations of $\varepsilon$ across the population. Of note, the regression line has a negative intercept and the data cloud seems displaced downwards compared to what one might expect. As a result, $\Delta\rho^{neuronal}$ was on average close to 0. We cannot provide a clear interpretation for this observation and future work shall revisit this issue.

The online version of this article includes the following figure supplement(s) for figure 5:

**Figure supplement 1.** The order bias does not reflect differences in the tuning of *offer value* cells.

In conclusion, the order bias did not originate before or during valuation. Analysis of *chosen juice* cells and *chosen value* cells indicated that the order bias emerged during value comparison (decision stage).

## The preference bias emerged late in the trial (post-comparison)

When offers were presented sequentially (Task 2), both monkeys showed an additional preference bias that favored juice A and was more pronounced when the relative value of the two juices was larger (*Figure 3*). Our last series of analyses focused on the origins of this bias.

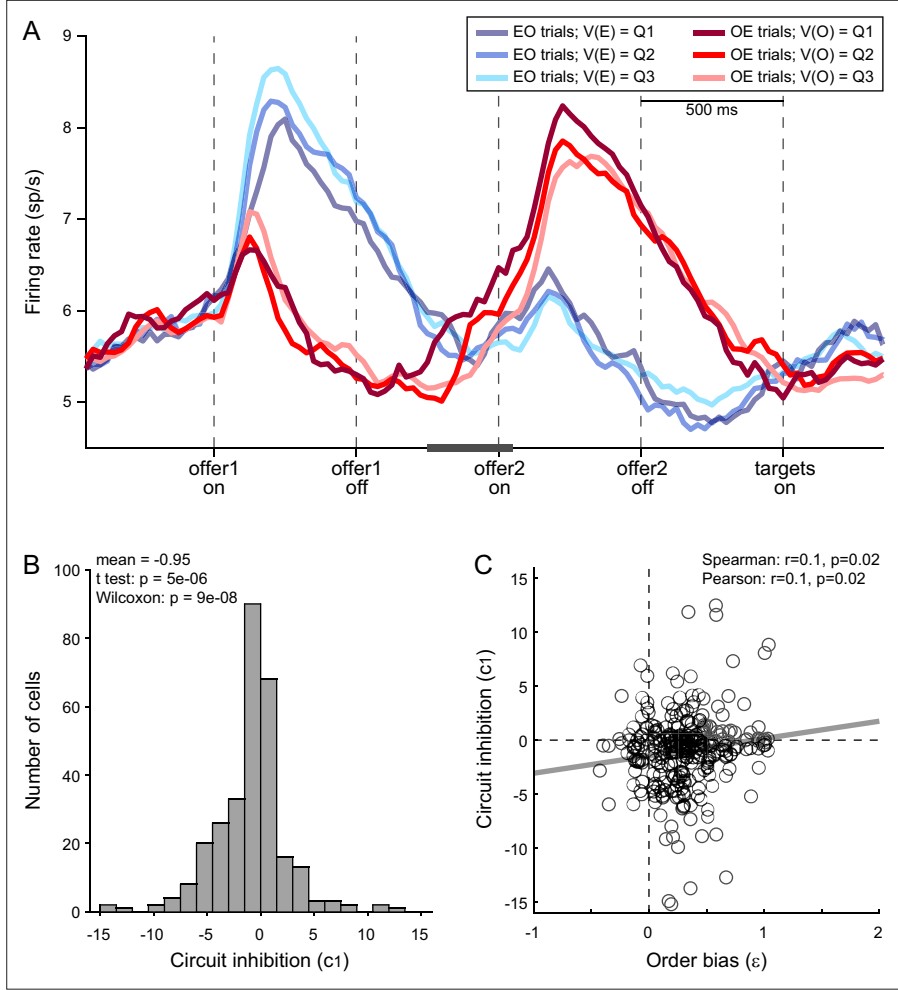

**Figure 6.** Order bias and circuit inhibition. (**A**) Circuit inhibition in *chosen juice* cells (primary data set, *N* = 160 cells). For each *chosen juice* cell E and O indicated the encoded juice and the other juice, respectively. We separated EO and OE trials, and divided each group of trials in tertiles based on the value of offer1. For EO trials, this corresponded to V(E); for OE trials, it corresponded to V(O). In this panel, Q1, Q2, and Q3 indicate low, medium, and high values of offer1. In OE trials, shortly before offer2, the activity of *chosen juice* cells was negatively correlated with V(O) – a phenomenon termed circuit inhibition. For a quantitative analysis of circuit inhibition, we focused on 300ms time window starting 250 ms before offer2 onset (black line). (**B**) Circuit inhibition for individual cells (*N* = 295 cells). For each *chosen juice* cell, we regressed the firing rate against the normalized V(O) (see Methods). The histogram illustrates the distribution of regression slopes ($c_1$), which quantify circuit inhibition for individual cells. The effect was statistically significant across the population (mean = −0.95). (**C**) Correlation between order bias and circuit inhibition (*N* = 295 cells). Here, the *x*-axis is the order bias ($\varepsilon$), the *y*-axis is circuit inhibition (regression slope $c_1$) and each data point represents one cell. The two measures were significantly correlated across the population. Panel A includes only the primary data set; thus circuit inhibition shown here replicates previous findings (***Ballesta and Padoa-Schioppa, 2019***). Panels B and C include both the primary and the additional data sets (see Methods). In panels B and C, 47 cells were excluded from the analysis because measures of order bias ($\varepsilon$) or circuit inhibition ($c_1$) were detected as outliers by the interquartile criterion. Including these cells in the analysis did not substantially alter the results. Statistical tests and exact p values are indicated in panels B and C.

First, we inquired whether the preference bias emerged during valuation. If this was the case, one or both of the following should be true: (1) *offer value A* cells encoded higher values in Task 2 than in Task 1 and/or (2) *offer value B* cells encoded lower values in Task 2 than in Task 1. Furthermore, these putative effects should increase as a function of the relative value. To test these predictions, we examined the tuning functions of *offer value* cells. For each cell group (*offer value A*, *offer value B*), we pooled neurons with positive and negative encoding. For Task 1, we focused on the post-offer

time window; for Task 2, we focused on post-offer1 and post-offer2 time windows, pooling trial types from both windows. Indicating with $b_0$ and $b_1$ the tuning intercept and tuning slope (see Methods, **Equation 8**), we computed the difference in intercept $\Delta b_0 = b_{0,Task2} - b_{0,Task1}$ and the difference in slope $\Delta b_1 = b_{1,Task2} - b_{1,Task1}$ for each cell. We then examined the relation between these measures and the relative value $\rho$ across the population, separately for each cell group. Contrary to the prediction, we did not find any correlation between neuronal measures ($\Delta b_0$, $\Delta b_1$) and the behavioral measure ($\rho$) for either *offer value A* or *offer value B* cells (**Figure 7—figure supplement 1**). Thus, the preference bias did not seem to emerge at the valuation stage.

We next examined *chosen value* cells. As discussed above, their activity provided a neuronal measure for the relative value ($\rho^{neuronal}$), which reflected the internal subjective values of the juices emerging during value comparison. In principle, $\rho^{neuronal}$ might differ from the relative value derived from choices through the probit regression ($\rho^{behavioral}$) because choices might be affected

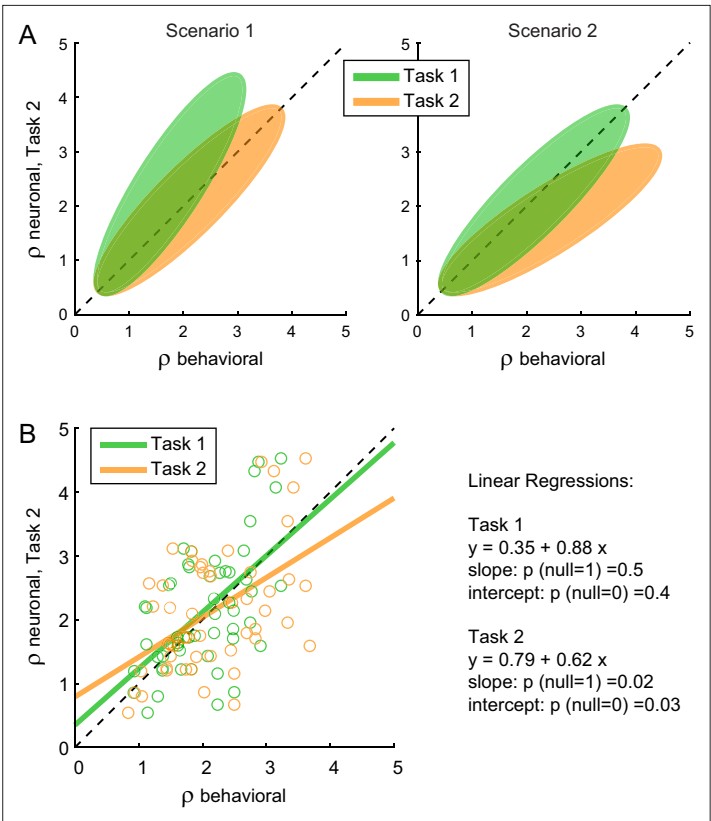

**Figure 7.** The preference bias does not reflect differences in the activity of *chosen value* cells. (**A**) Hypothetical scenarios. The two panels represent in cartoon format two possible scenarios envisioned at the outset of this analysis. In both panels, the *x*-axis represents behavioral measures from either Task 1 (green) or Task 2 (yellow); the *y*-axis represents the neuronal measure from Task 2. In scenario 1, the animal assigned higher relative value to juice A in Task 2. Thus, neuronal measures of relative value derived from the activity of *chosen value* cells in Task 2 ($\rho^{neuronal}_{Task2}$) would align with behavioral measures from the same task ($\rho^{behavioral}_{Task2}$) and be systematically higher than behavioral measures from Task 1 ($\rho^{behavioral}_{Task1}$). In scenario 2, the animal assigned the same relative values to the juices in both tasks. Thus, neuronal measures of relative value in Task 2 ($\rho^{neuronal}_{Task2}$) would be systematically lower than behavioral measures from the same task ($\rho^{behavioral}_{Task2}$) and would align with behavioral measures from Task 1 ($\rho^{behavioral}_{Task1}$). (**B**) Empirical results ($N = 52$ cells). Neuronal measures derived from Task 2 ($\rho^{neuronal}_{Task2}$) are plotted against behavioral measures obtained in Task 1 ($\rho^{behavioral}_{Task1}$, green) or Task 2 ($\rho^{behavioral}_{Task2}$), yellow. Lines are from linear regressions. In essence, $\rho^{neuronal}_{Task2}$ was statistically indistinguishable from $\rho^{behavioral}_{Task1}$ and systematically lower than $\rho^{behavioral}_{Task2}$. Details on the statistics and exact p values are indicated in the figure. The analysis was restricted to 52 cells that had significant $\theta_{1A}$, $\theta_{1B}$, $\theta_{2A}$, and $\theta_{2B}$. For this analysis, $\rho^{neuronal}_{Task2}$ was taken as equal to $\rho^{neuronal}_{offer2}$ (**Equation 14**). Other definitions provided similar results (data not shown).

The online version of this article includes the following figure supplement(s) for figure 7:

**Figure supplement 1.** The preference bias does not reflect differences in the tuning of *offer value* cells.

by systematic biases originating downstream of value comparison (*Figure 1*). In the light of this consideration, we examined the relation between the neuronal measure of relative value in Task 2 ($\rho^{neuronal}_{Task2}$, see Methods) and the behavioral measures obtained in the two tasks ($\rho^{behavioral}_{Task1}$, $\rho^{behavioral}_{Task2}$). We envisioned two possible scenarios (*Figure 7A*). In scenario 1, the preference bias reflected a difference in values across tasks. In other words, the subjective values of the juices in the two tasks were different and such that the relative value of juice A was higher in Task 2 than in Task 1. If so, $\rho^{neuronal}_{Task2}$ should be statistically indistinguishable from $\rho^{behavioral}_{Task2}$ and systematically larger than $\rho^{behavioral}_{Task1}$. In scenario 2, the subjective values of the juices were the same in both tasks and the preference bias reflected some neuronal process taking place downstream of value comparison. If so, $\rho^{neuronal}_{Task2}$ should be statistically indistinguishable from $\rho^{behavioral}_{Task1}$ and systematically smaller than $\rho^{behavioral}_{Task2}$.

The results of our analysis clearly conformed with scenario 2 (*Figure 7B*). For each *chosen value* cell, we computed $\rho^{neuronal}_{Task1}$ in the post-offer time window and $\rho^{neuronal}_{Task2}$ in the post-offer2 time window. Across the population, the two measures were statistically indistinguishable (p = 0.3, *t*-test; not shown). We then regressed $\rho^{neuronal}_{Task2}$ onto $\rho^{behavioral}_{Task1}$. The linear relation between these measures was statistically indistinguishable from identity. Separately, we regressed $\rho^{neuronal}_{Task2}$ onto $\rho^{behavioral}_{Task2}$. In this case, the regression slope was significantly <1 (p = 0.02). This result is quite remarkable. It shows that the chosen value represented in the brain in Task 2 was equal to that inferred from choices in Task 1, and significantly different from that inferred from choices in Task 2. This fact implies that the preference bias was costly for the monkey, as it reduced the value obtained on average at the end of each trial (see Discussion).

In summary, the preference bias did not reflect differences in the values assigned to individual offers (offer values). Furthermore, insofar as the activity of *chosen value* cells reflects the decision process (*Figure 1*), the preference bias did not seem to emerge during value comparison. So how can one make sense of this behavioral phenomenon? At the cognitive level, the preference bias might be interpreted as due to the higher demands of Task 2. When the two saccade targets appeared on the monitor, information about values was no longer on display (*Figure 2B*). If at that point the animal had not finalized its decision, or if it had failed to retain in working memory the decision outcome, the animal might have selected the target associated with the better juice (juice A). Such bias would have been especially strong when the value difference between the two juices was large. In this view, the preference bias would reflect a 'second thought' occurring after value comparison, in some trials.

To test this intuition, we turned to the activity of *chosen juice* cells. As noted above, in Task 2, the evolving decision was captured by the activity of these neurons recorded in OE trials immediately before and after offer2 onset (*Figure 8A*). More specifically, the state of the ongoing decision was captured by the distance between the two traces corresponding to the two possible choice outcomes (E chosen, O chosen). For any neuron, we quantified this distance with a receiver operating characteristic (ROC) analysis, which provided a choice probability (CP). In essence, CP can be interpreted as the probability with which an ideal observer may guess the eventual choice outcome based on the activity of the cell. For each *chosen juice* cell, we computed the CP at different times in the trial. Across the population, mean(CP) exceeded chance level starting shortly before offer2, consistent with the above discussion on circuit inhibition. We then proceeded to investigate the origins of the preference bias.

We reasoned that, at the net of noise in measurements and cell-to-cell variability, CPs ultimately quantify the animal's commitment to the eventual choice outcome. If the preference bias emerged late in the trial – perhaps after target presentation, if animals had not already finalized their decision – the intensity of the preference bias should be inversely related to the animals' commitment to the eventual choice outcome measured earlier in the trial. In other words, there should be a negative correlation between the preference bias and CPs computed at the time when decisions normally take place (shortly before or after offer2 onset). Our analyses supported this prediction. To quantify the preference bias intensity independent of the juice pair, we defined the preference bias index (PBI) = 2 $(\rho_{Task2} - \rho_{Task1})/(\rho_{Task2} + \rho_{Task1})$. We then focused on four 250-ms time windows before offer1 (control window), before and after offer2 onset, and before juice delivery (*Figure 8B–E*). Confirming our predictions, CP and PBI were significantly anti-correlated immediately before and during offer2 presentation, but not in the control time window or late in the trial (*Figure 8F–I*).

In conclusion, our results indicated that the preference bias did not emerge during valuation or during value comparison. Conversely, our results suggest that the preference bias emerged late in the

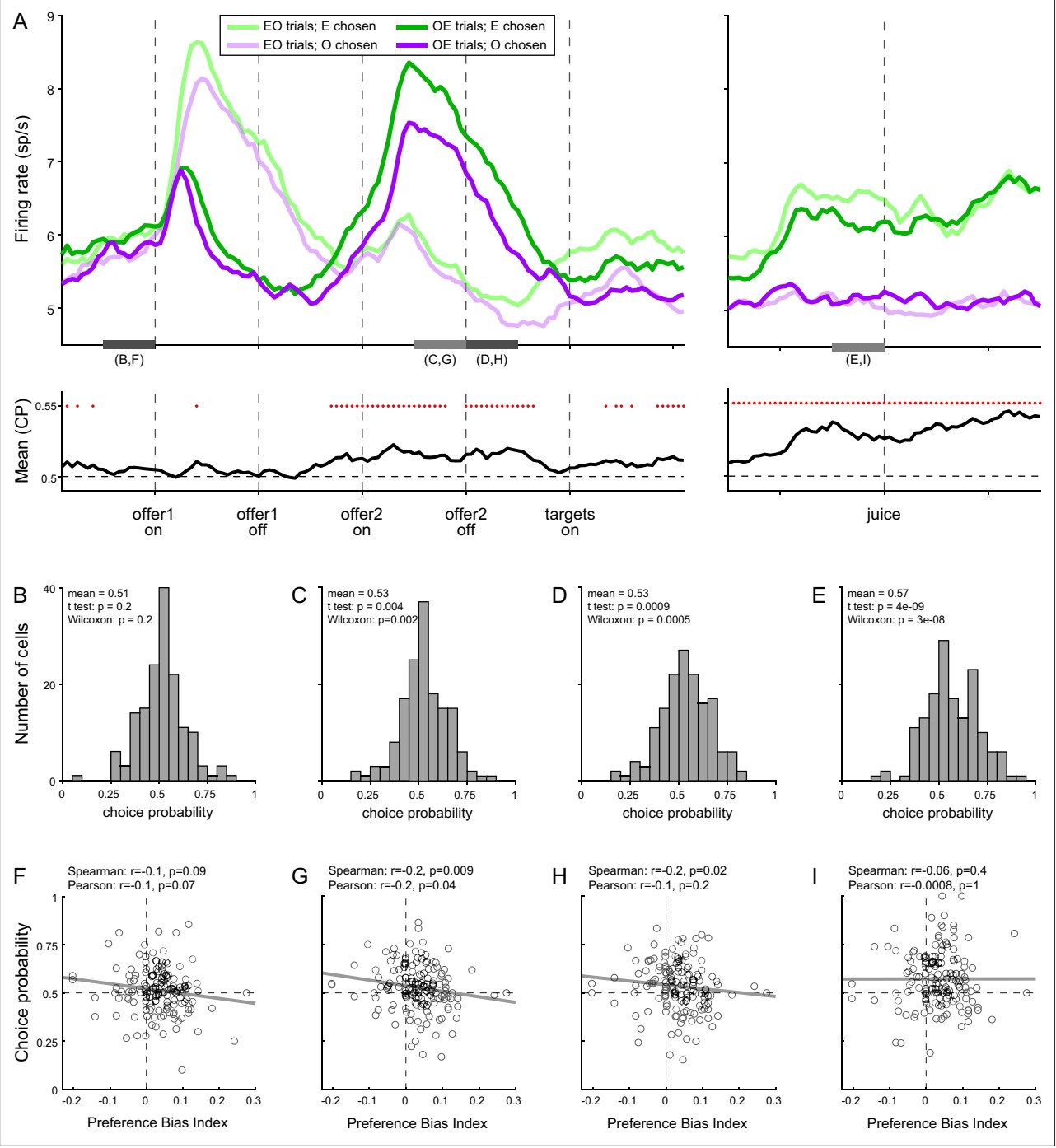

**Figure 8.** Preference bias and choice probability (CP) in *chosen juice* cells. (**A**) Profiles of activity and CP (*N* = 160 cells). On the top, separate traces are activity profiles for EO trials (dark colors) and OE trials (light colors), separately for E chosen (blue) and O chosen. On the bottom the trace is the mean(CP) computed for OE trials in 100-ms sliding time windows (25-ms steps). Red dots indicate that mean(CP) was significantly >0.5 (p < 0.001; *t*-test). Value comparison typically takes place shortly after the onset of offer2. (**B–E**) Distribution of CP in four 250-ms time windows. The time windows used for this analysis are indicated in panel A. (**F–I**) Correlation between CP and preference bias index. Each panel corresponds to the histogram immediately above it. CPs are plotted against the preference bias index (PBI), which quantifies the preference bias independently of the juice types. Each symbol represents one cell and the line is from a linear regression. CP and PBI were negatively correlated immediately before and after offer2 onset, but not later in the trial. This pattern suggests that the preference bias emerged late in the trial, when decisions were not finalized shortly after offer2 presentation.

trial, as a 'second thought' process that guided choices when decisions were not finalized based on offer values alone.

## Discussion

### Behavioral values, neuronal values, and the origins of choice biases

Early economists proposed that choices between goods entail the computation and comparison of subjective values. However, the concept of value is somewhat slippery, because values relevant to choices cannot be measured behaviorally other than from choices themselves. This circularity problem haunted generations of scholars, dominating academic debates in the 19th and 20th centuries. In the end, neoclassic economic theory came to reject (cardinal) values and to rely only on (ordinal) preferences (*Niehans, 1990*; *Samuelson, 1947*). In other words, standard economics is agnostic as to whether subjective values are computed at all. The construction of standard economic theory was a historic success, but it came at a cost: the theory cannot explain a variety of biases observed in human choices (*Camerer et al., 2003*; *Kahneman and Tversky, 2000*; *Lichtenstein and Slovic, 2006*). In this perspective, neuroscience results showing that neuronal activity in multiple brain regions is linearly related to values defined behaviorally (*Amemori and Graybiel, 2012*; *Cai et al., 2011*; *Cai and Padoa-Schioppa, 2012*; *Hosokawa et al., 2013*; *Jezzini and Padoa-Schioppa, 2020*; *Kable and Glimcher, 2007*; *Kim et al., 2008*; *Lak et al., 2014*; *Levy et al., 2010*; *Louie and Glimcher, 2010*; *Padoa-Schioppa and Assad, 2006*; *Pastor-Bernier et al., 2019*; *Plassmann et al., 2007*; *Shenhav and Greene, 2010*), constitute a significant breakthrough. They validate the concept of value and effectively break the circularity surrounding it. Indeed, a neuronal population whose activity is reliably correlated with values measured from choices (behavioral values) may be used to derive independent measures of subjective values (neuronal values). In most circumstances, neuronal values and behavioral values should be (and are) indistinguishable. However, in specific choice contexts, the two measures might differ somewhat. When observed, such discrepancies indicate that choices are partly determined by processes that escape the maximization of offer values. If so, suitable analyses of neuronal activity may be used to assess the origins of particular choice biases.

These considerations motivated the analyses conducted in this study. In our experiments, monkeys chose between two juices offered simultaneously or sequentially. Choices under sequential offers were less accurate, biased in favor of the second offer (order bias), and biased in favor of the preferred juice (preference bias). It is generally understood that good-based economic decisions take place in OFC (*Cisek, 2012*; *Padoa-Schioppa, 2011*; *Rushworth et al., 2012*) and that the encoding of decision variables in this area is categorical in nature (*Hirokawa et al., 2019*; *Onken et al., 2019*; *Padoa-Schioppa, 2013*). Earlier studies had identified in OFC three groups of neurons encoding individual offer values, the chosen juice and the chosen value. Furthermore, choices under simultaneous or sequential offers were found to engage the same groups of cells (*Shi et al., 2022a*). Notably, the variables encoded in OFC capture both the input and the output of the decision process. This observation and a series of experimental and theoretical results lead to the hypothesis that the cell groups identified in OFC constitute the building blocks of a decision circuit (*Padoa-Schioppa and Conen, 2017*). More specifically, we hypothesize that *offer value* cells provide the primary input to a circuit formed by *chosen juice* cells and *chosen value* cells, where decisions are formed. In this view, different cell groups in OFC may be associated with different computational stages: *offer value* cells instantiate the valuation stage; *chosen value* cells reflect values possibly modified by the decision process; and *chosen juice* cells capture the evolving commitment to a particular choice outcome. In this framework, we examined the activity of each cell group in relation to each behavioral phenomenon.

Our results may be summarized as follows. (1) Other things equal, neuronal signals encoding the offer values were weaker (smaller activity range) under sequential offers than under simultaneous offers. The reason for this discrepancy is unclear, but this neuronal effect was correlated with the difference in choice accuracy measured at the behavioral level. In other words, the drop in choice accuracy observed under sequential offers originated, at least partly, at the valuation stage. (2) The order bias did not correlate with any measure in the activity of *offer value* cells. However, the order bias was negatively correlated with circuit inhibition in *chosen juice* cells – a phenomenon seen as key to value comparison (*Ballesta and Padoa-Schioppa, 2019*). Furthermore, session-to-session fluctuations in the order bias correlated with fluctuations in the neuronal measure of relative value derived

from *chosen value* cells. These findings indicate that the order bias emerged during value comparison. (3) The preference bias did not have any correlate in the activity of *offer value* cells or *chosen value* cells. Moreover, the preference bias was inversely related to a measure derived from *chosen juice* cells and quantifying the degree to which the decision was finalized when offer values are 'normally' compared (i.e., upon presentation of the second offer). These findings indicate that the preference bias emerged late in the trial. As a caveat, the hypothesis discussed above, linking different cell groups in OFC to specific decision stages, awaits further confirmation.

Two of our findings are particularly relevant to the distinction between behavioral values and neuronal values. First, the activity of *offer value* cells did not present any difference associated with the presentation order or with the juice preference. Second, relative values derived from *chosen value* cells under sequential offers differed significantly from behavioral measures obtained in the same task, and were indistinguishable from behavioral measures obtained in the other task (simultaneous offers). Thus, the order bias and the preference bias highlighted significant differences between neuronal and behavioral measures of value. These observations imply that the order bias and the preference bias emerged downstream of valuation. Importantly, they also imply that the two choice biases imposed a cost to the animals, in the sense that they reduced the (neuronal) value obtained on average in any given trial. Notably, it would be impossible to draw such conclusion based on choices alone.

To our knowledge, this is the first study to investigate the origins of choice biases building on the distinction between behavioral values and neuronal values. At the same time, some of our results are not unprecedented. Earlier work showed that human and animal choices are affected by a bias favoring, on any given trial, the same good chosen in the previous trial (*Alós-Ferrer et al., 2016*; *Goodwin, 1977*; *Padoa-Schioppa, 2013*; *Schoemann and Scherbaum, 2019*; *Senftleben et al., 2021*). The origins of this phenomenon, termed choice hysteresis, are hard to pinpoint based on behavioral evidence alone. However, previous analysis of neuronal activity in OFC revealed that choice hysteresis is not reflected in the encoding of offer values. Conversely, choice hysteresis correlates with fluctuations in the baseline activity of *chosen juice* cells, which is partly influenced by the previous trial's outcome (*Padoa-Schioppa, 2013*). Thus, similar to the order bias, choice hysteresis appears to emerge at the decision stage.

## The cost of choice biases

We have noted that the three behavioral phenomena described here were detrimental to the animals. This point bears a few comments.

Let us consider, in a very general sense, choices between two goods A and B, taking place in two possible conditions 1 and 2. We may refer to the subjective values of the two goods in the two conditions as $V_{A,1}$, $V_{A,2}$, $V_{B,1}$, and $V_{B,2}$. Lets also assume the presence of a choice bias such that in condition 1 the animal consistently chooses A, while in condition 2 it consistently chooses B. In broad strokes, that might be for two reasons. Either (1) values differ across conditions such that $V_{A,1} > V_{B,1}$ and $V_{A,2} < V_{B,2}$, or (2) the value of each good remains unchanged across conditions ($V_{A,1} = V_{A,2}$ and $V_{B,1} = V_{B,2}$) and choices are affected by some other process, downstream of value computation. If so, in one of the two conditions the animal consistently chooses the lower value. In other words, the choice bias is detrimental to the animal. Coming to our experiments, the analysis of neuronal activity indicates that the subjective values of offered juices were independent of the choice task, and independent of the presentation order in Task 2. Thus, scenario (2) held true with respect to both the order bias and the preference bias. Consequently, both biases were detrimental to the animals.

With respect to the preference bias, one question is whether the bias affected choices in Task 1 or in Task 2 (in principle, there could be a bias favoring the unpreferred juice in Task 1). The fact that $\rho^{behavioral}_{Task1}$, $\rho^{neuronal}_{Task1}$, and $\rho^{neuronal}_{Task2}$ were all indistinguishable from each other while $\rho^{behavioral}_{Task2}$ differed from them (*Figure 7*) argues for the latter understanding.

It is interesting to speculate whether the choice biases documented here might benefit the animal in some more general sense. For example, one might wonder whether the cost imposed by the preference bias was lower than the metabolic cost the monkey would have incurred to increase its performance level and avoid that bias. If so, the preference bias would be, in fact, ecologically adaptive. Addressing this question would require quantifying the metabolic cost of increasing performance in the same value units used for the juices – a challenge open for future studies. However, independent of that assessment, our present results indicate that the putative metabolic cost of increasing

performance in the task did not explicitly enter the decision process. If metabolic cost affected behavior, it did so in a meta-decision sense.

Similar considerations hold for the difference in choice accuracy measured across tasks. The fact that sigmoid functions are not infinitely steep (i.e., the presence of choice variability) means that on some trials the animal chooses the lower value. In fact, one can quantify the loss in expected payoff as a function of the sigmoid steepness (*Constantinople et al., 2019*; *Rustichini et al., 2017*). That sigmoids were shallower in Task 2 means that the average payoff was lower in that task – a detriment to the animal. Again, it is interesting to speculate whether weaker offer value signals recorded in Task 2 might also benefit the animal in some way, perhaps by reducing cognitive or metabolic costs. This question remains open for future studies. Importantly, such costs did not explicitly enter the decision process; if they affected behavior, they did so in a meta-decision sense.

## Conclusions

The past two decades have witnessed a lively interest for the neural underpinnings of choice behavior. In this effort, a significant breakthrough came from the adoption of behavioral paradigms inspired by the economics literature, in which subjective values derived from choices are used to interpret neural activity. Without renouncing this approach, here we took a further step, showing that the decision process sometimes falls short of selecting the maximum offer value, and that choices are sometimes affected by processes taking place downstream of value comparison. In other words, behavioral values and neuronal values sometimes differ. These results might seem uncontroversial, but they have deep implications for economic theory and beyond. Looking forward, the framework developed here, in which the computation and comparison of offer values are central, but choices can also be affected by other processes accessible through neuronal measures, may help understand the origins of other choice biases.

# Materials and methods

All the experimental procedures adhered to the *NIH Guide for the Care and Use of Laboratory Animals* and were approved by the Institutional Animal Care and Use Committee (IACUC) at Washington University (protocol number 190931).

## Animal subjects, choice tasks, and neuronal recordings

This study presents new analyses of published data (*Shi et al., 2022a*). Experimental procedures for surgery, behavioral control, and neuronal recordings have been described in detail. Briefly, two male monkeys (*Macaca mulatta*; monkey J, 10.0 kg, 8 years old; monkey G, 9.1 kg, 9 years old) participated in the study. Under general anesthesia, we implanted on each animal a head restraining device and an oval chamber (axes 50 × 30 mm) allowing bilateral access to OFC. During the experiments, monkeys sat in an electrically insulated environment with their head fixed and a computer monitor placed at 57 cm distance. The gaze direction was monitored at 1 kHz using an infrared video camera (Eyelink, SR Research). Behavioral tasks were controlled through custom written software based on Matlab (v2016a; MathWorks Inc). The code is available online (*Hwang et al., 2019*; *Shi et al., 2022b*; https://monkeylogic.nimh.nih.gov).

In each session, the animal chose between two juices labeled A and B (A preferred) offered in variable amounts. Trials with two choice tasks, referred to as Task 1 and Task 2, were pseudorandomly interleaved. In both tasks, offers were represented by sets of colored squares displayed on the monitor. For each offer, the color indicated the juice type and the number of squares indicated the quantity. Each trial began with the animal fixating a large dot. After 0.5 s, the initial fixation point changed to a small dot or a small cross; the new fixation point cued the animal to the choice task used in that trial. In Task 1 (*Figure 2A*), cue fixation (0.5 s) was followed by the simultaneous presentation of the two offers. After a randomly variable delay (1–1.5 s), the center fixation point disappeared and two saccade targets appeared near the offers (go signal). The animal indicated its choice with an eye movement. It maintained peripheral fixation for 0.75 s, after which the chosen juice was delivered. In Task 2 (*Figure 2B*), cue fixation (0.5 s) was followed by the presentation of one offer (0.5 s), an interoffer delay (0.5 s), presentation of the other offer (0.5 s), and a wait period (0.5 s). Two colored saccade targets then appeared on the two sides of the fixation point. After a randomly variable delay

(0.5–1 s), the center fixation point disappeared (go signal). The animal indicated its choice with a saccade, maintained peripheral fixation for 0.75 s, after which the chosen juice was delivered. Central and peripheral fixation were imposed within 4–6 and 5–7 degrees of visual angle, respectively. Aside from the initial cue, the choice tasks were nearly identical to those used in previous studies (*Ballesta and Padoa-Schioppa, 2019*; *Padoa-Schioppa and Assad, 2006*).

For any given trial, $q_A$ and $q_B$ indicate the quantities of juices A and B offered to the animal, respectively. An 'offer type' was defined by two quantities [$q_A$ $q_B$]. On any given session, we used the same juices and the same sets of offer types for the two tasks. For Task 1, the spatial configuration of the offers varied randomly from trial to trial. For Task 2, the presentation order varied pseudorandomly and was counterbalanced across trials for any offer type. The terms 'offer1' and 'offer2' indicated, respectively, the first and second offer, independently of the juice type and amount. Trials in which juice A was offered first and trials in which juice B was offered first were referred as 'AB trials' and 'BA trials', respectively. The spatial location (left/right) of saccade targets varied randomly. The juice volume corresponding to one square (quantum) was set equal for the two choice tasks and remained constant within each session. It varied across sessions between 70 and 100 µl for both monkeys. The association between the initial cue (small dot, small cross) and the choice task varied across sessions in blocks. Across sessions, we used 12 different juices (and colors) and 45 different juice pairs. Based on a power analysis, in most sessions the number of trials for Task 2 was set equal to 1.5 times that for Task 1.

Neuronal recordings were guided by structural MRI scans (1 mm sections) obtained before and after surgery and targeted area 13 m (*Ongür and Price, 2000*). We recorded from both hemispheres in both monkeys. Tungsten single electrodes (100 µm shank diameter; FHC) were advanced remotely using a custom-built motorized microdrive. Typically, one motor advanced two electrodes placed 1 mm apart, and 1–2 such pairs of electrodes were advanced unilaterally or bilaterally in each session. Neural signals were amplified (gain: 10,000) bandpass filtered (300 Hz to 6 kHz; Lynx 8, Neuralynx), digitized (frequency: 40 kHz) and saved to disk (Power 1401, Cambridge Electronic Design). Spike sorting was performed offline (Spike2, v6, Cambridge Electronic Design). Only cells that appeared well isolated and stable throughout the session were included in the analysis.

## Behavioral analyses

In each session, choice patterns were analyzed using probit regressions as described in the main text (*Equations 1 and 2*). For convenience, we repeat here the equation only for Task 1.

$$Choice\ B = \Phi(X)$$
$$X = a_0 + a_1\ log(q_B/q_A) \tag{4}$$

Here, $\Phi$ indicates the cumulative function of the standard normal distribution. This model is referred to as the 'log value ratio' model. For Task 1 (simultaneous offers), the probit fit provided measures for the relative value $\rho_{Task1}$ and the sigmoid steepness $\eta_{Task1}$. For Task 2 (sequential offers), the probit fit provided measures for the relative value $\rho_{Task2}$, the sigmoid steepness $\eta_{Task2}$, and the order bias $\varepsilon$. Subsequent analyses of neuronal activity relied on these behavioral measures.

To test the robustness of our findings, we conducted a series of control analyses. First, we fitted a probit using a 'value difference' model, defined as follows:

$$Choice\ B = \Phi(X)$$
$$X = a_0\ q_A + a_1\ q_B \tag{5}$$

Second, we fitted a logit using a log value ratio model:

$$Choice\ B = 1/(1 + e^{-X})$$
$$X = a_0 + a_1\ log(q_B/q_A) \tag{6}$$

Third, we fitted a logit using a value difference model:

$$Choice\ B = 1/(1 + e^{-X})$$
$$X = a_0\ q_A + a_1\ q_B \tag{7}$$

Each of these fit provided measures for each of the parameters characterizing choices in the two tasks ($\rho_{Task1}$, $\rho_{Task2}$, etc.). For each session and for each model we obtained an $R^2$. We then compared different models by computing the distribution of BIC across sessions for each pair of models. We generally found that log value ratio models provided a better fit compared to value difference models, consistent with theoretical considerations (*Padoa-Schioppa, 2022*). We also found that logit models provided a better fit compared to probit models, although measures of relative value, sigmoid steepness, and order bias were very similar and highly correlated. For consistency with previous studies, we report the results of neuronal analyses based on neuronal measures derived from *Equations 1 and 2*. However, all our results held true using measures derived from logit regressions.

Notably, *Equation 2* describes two parallel sigmoids. In a control analysis, we relaxed this assumption and fitted choices in AB and BA trials with two independent sigmoids. Analyzing neuronal activity based on measures derived from this analysis did not substantially alter any of the results.

Finally, we defined the order bias as $\varepsilon = 2\,\rho_{Task2}\,a_4/a_3$. This definition is particularly convenient for the present analyses because ε equals the difference $\rho_{BA} - \rho_{AB}$ (*Equation 3*). Alternative and valid definitions include $\varepsilon = a_4$ and $\varepsilon = a_4/a_3$. Control analyses showed that using these definitions did not substantially alter any of the results.

## Preliminary analyses of neuronal activity

The present analyses build on the results of a previous study showing that both choice tasks engage the same groups of neurons in OFC (*Shi et al., 2022a*). Here, we briefly summarize those findings.

The original data set included 1526 neurons (672 from monkey J, 854 from monkey G) recorded in 306 sessions (115 from monkey J, 191 from monkey G). For each neuron, trials from Task 1 and Task 2 were first analyzed separately using the procedures developed in previous studies (*Ballesta and Padoa-Schioppa, 2019*; *Padoa-Schioppa and Assad, 2006*). For Task 1, we defined four time windows: post-offer (0.5 s after offer onset), late-delay (0.5–1 s after offer onset), pre-juice (0.5 s before juice onset), and post-juice (0.5 s after juice onset). A 'trial type' was defined by two offered quantities and a choice. For Task 2, we defined three time windows: post-offer1 (0.5 s after offer1 onset), post-offer2 (0.5 s after offer2 onset) and post-juice (0.5 s after juice onset). A 'trial type' was defined by two offered quantities, their order and a choice. For each task, each trial type and each time window, we averaged spike counts across trials. A 'neuronal response' was defined as the firing rate of one cell in one time window as a function of the trial type. Neuronal responses in each task were submitted to an analysis of variance (factor: trial type). Neurons passing the $p < 0.01$ criterion in ≥1 time window in either task were identified as 'task-related' and included in subsequent analyses.

Following earlier work (*Padoa-Schioppa, 2013*), neurons in Task 1 were classified in one of four groups *offer value A*, *offer value B*, *chosen juice*, or *chosen value*. Each variable could be encoded with positive or negative sign, leading to a total of eight cell groups. Each neuronal response was regressed against each of the four variables. If the regression slope $b_1$ differed significantly from zero ($p < 0.05$), the variable was said to 'explain' the response. In this case, we set the signed $R^2$ as $sR^2 = \text{sign}(b_1)\,R^2$; if the variable did not explain the response, we set $sR^2 = 0$. After repeating the operation for each time window, we computed for each cell the $sum(sR^2)$ across time windows. Neurons explained by at least one variable in one time window, such that $sum(sR^2) \neq 0$, were said to be tuned; other neurons were labeled 'untuned'. Tuned cells were assigned to the variable and sign providing the maximum $|sum(sR^2)|$, where $|\cdot|$ indicates the absolute value. Thus, indicating with '+' and '−' the sign of the encoding, each neuron was classified in one of nine groups: *offer value A+*, *offer value A−*, *offer value B+*, *offer value B−*, *chosen juice A*, *chosen juice B*, *chosen value+*, *chosen value−*, and *untuned*.

Neuronal classification in Task 2 followed the procedures described in a previous study (*Ballesta and Padoa-Schioppa, 2019*). Under sequential offers, neuronal responses in OFC were found to encode different variables defined in relation to the presentation order (AB or BA). Specifically, the vast majority of responses were explained by one of 11 variables including 1 binary variable capturing the presentation order (*AB | BA*), 6 variables representing individual offer values (*offer value A | AB*, *offer value A | BA*, *offer value B | AB*, *offer value B | BA*, *offer value 1*, and *offer value 2*), 3 variables capturing variants of the chosen value (*chosen value*, *chosen value A*, and *chosen value B*), and a binary variable representing the binary choice outcome (*chosen juice*). Each of these variables could be encoded with a positive or negative sign. Most neurons encoded different variables in different

time windows. In principle, considering 11 variables, 2 signs of the encoding and 3 time windows, neurons might present a very large number of variable patterns across time windows. However, the vast majority of neurons presented one of eight patterns referred to as 'sequences'. Classification proceeded as follows. For each cell and each time window, we regressed the neuronal response against each of the variables predicted by each sequence. If the regression slope $b_1$ differed significantly from zero (p < 0.05), the variable was said to explain the response and we set the signed $R^2$ as $sR^2 = sign(b_1) R^2$; if the variable did not explain the response, we set $sR^2 = 0$. After repeating the operation for each time window, we computed for each cell the $sum(sR^2)$ across time windows for each of the eight sequences. Neurons such that $sum(sR^2) \neq 0$ for at least one sequence were said to be tuned; other neurons were untuned. Tuned cells were assigned to the sequence that provided the maximum $|sum(sR^2)|$. As a result, each neuron was classified in one of nine groups: *seq #1*, *seq #2*, *seq #3*, *seq #4*, *seq #5*, *seq #6*, *seq #7*, *seq #8*, and *untuned* (**Table 1**).

The results of the two classifications were compared using analyses for categorical data. In essence, we found a strong correspondence between the cell classes identified in the two choice tasks (**Shi et al., 2022a**). Hence, we may refer to the different groups of cells using the standard nomenclature – *offer value*, *chosen juice*, and *chosen value* – independently of the choice task. Based on this result, we proceeded with a comprehensive classification based on the activity recorded in both choice tasks. For each task-related cell, we calculated the $sum(sR^2)$ for the eight variables in Task 1 ($sum(sR^2)_{Task1}$) and eight sequences in Task 2 ($sum(sR^2)_{Task2}$) as described above. We then added the corresponding $sum(sR^2)_{Task1}$ and $sum(sR^2)_{Task2}$ to obtain the final $sum(sR^2)_{final}$. Neurons such that $sum(sR^2)_{final} \neq 0$ for at least one class were said to be tuned; other neurons were untuned. Tuned cells were assigned to the cell class that provided the maximum $|sum(sR^2)_{final}|$.

## Data sets

In some sessions, one or both choice patterns presented complete or quasicomplete separation – that is, the animal split choices for <2 offer types in Task 1 and/or in Task 2. In these cases, the probit regression did not converge, the resulting steepness $\eta$ was high and unstable, and the relative value was not unique. This issue affected the classification analyses described above only marginally, but for the present study it was critical that behavioral measures be accurate and precise. We thus restricted our analyses to stable sessions by imposing an interquartile criterion on the sigmoid steepness (**Tukey, 1977**). Defining IQR as the interquartile range, values below the first quartile minus 1.5*IQR or above the third quartile plus 1.5*IQR were identified as outliers and excluded. Thus, our entire data set included 1204 neurons (577 from monkey J, 627 from monkey G) recorded in 241 sessions (101 from monkey J, 140 from monkey G). In this population, the classification procedures identified 183 *offer value* cells, 160 *chosen juice* cells, and 174 *chosen value* cells. These neurons constitute the primary data set for this study.

Most of our analyses compared choices and neuronal activity across tasks and were restricted to the primary data set. However, some analyses included only trials from Task 2 and quantified the effects due to the presentation order (AB vs. BA). In these analyses, we included an additional data set recorded previously from the same two animals performing only Task 2 (**Ballesta and Padoa-Schioppa, 2019**). All the procedures for behavioral control and neuronal recording were essentially identical to those described above. Furthermore, behavioral analyses and inclusion criteria were identical to those used for the primary data set. The resulting data set included 1205 neurons (414 from monkey J, 791 from monkey G) recorded in 196 sessions (51 from monkey J, 145 from monkey G). In this population, the classification procedures identified 243 *offer value* cells, 182 *chosen juice* cells, and 187 *chosen value* cells. We refer to these neurons as the additional data set. Importantly, the order bias was also observed in these sessions (**Ballesta and Padoa-Schioppa, 2019**).

The interquartile criterion was also used to identify outliers in all the analyses conducted throughout this study. In practice, this criterion became relevant only for the analyses shown in **Figure 6** and **Figure 5—figure supplement 1**, as indicated in the respective figure legends.

## Comparing tuning functions across choice tasks

Several analyses compared the tuning functions recorded in the two tasks. Tuning functions were defined by the linear regression of the firing rate *r* onto the encoded variable *S*:

$$r = b_0 + b_1 S \tag{8}$$

Regression coefficients $b_0$ and $b_1$ were referred to as tuning intercept and tuning slope, respectively. Positive and negative encoding corresponded to $b_1 > 0$ and $b_1 < 0$, respectively. We also defined the mean activity and the activity range as follows. Indicating with $S_{max}$ the maximum value of $S$, the mean activity was defined as $r_{mean} = b_0 + b_1 S_{max}/2$. The activity range was defined as $\Delta r = |b_1 S_{max}|$, where $|\cdot|$ indicates the absolute value.

For any neuronal response, the tuning was considered significant if $b_1$ differed significantly from zero ($p < 0.05$) and if the sign of the encoding was consistent with the cell class (e.g., $b_1 > 0$ for *offer value A +* cells). All the analyses comparing tuning functions across tasks were restricted to neuronal responses with significant tuning.

## Neuronal measures of relative value

Several analyses relied on neuronal measures for the relative value of the juices ($\rho^{neuronal}$) derived from the activity of *chosen value* cells. In Task 1, these neurons encode the *chosen value* independently of the juice type. For each neuronal response, we performed a bilinear regression:

$$r = \theta_0 + \theta_A\, q_A\, \delta_{choice,A} + \theta_B\, q_B\, \delta_{choice,B} \tag{9}$$

where $\theta_0$, $\theta_A$, and $\theta_B$ were the regression coefficients, $\delta_{choice,A} = 1$ if the animal chose juice A and 0 otherwise, and $\delta_{choice,B} = 1\ \delta_{choice,A}$. If the response encodes the *chosen value*, $\theta_A$ should be proportional to the value of a quantum of juice A ($u_A$), $\theta_B$ should be proportional to the value of a quantum of juice B ($u_B$), and the ratio $\theta_A/\theta_B$ should equal the value ratio – that is, the relative value of the two juices. We thus defined

$$\rho^{neuronal} = \theta_A/\theta_B \tag{10}$$

Previous studies showed that this measure is statistically indistinguishable from the behavioral measure $\rho^{behavioral}$ derived from the probit analysis of choice patterns (***Padoa-Schioppa and Assad, 2006***).

In Task 2, in the post-offer1 and post-offer2 time windows, *chosen value* cells encoded the value of the current offer, independent of the juice type (***Table 1***). For each neuron, we thus performed a bilinear regression for each of the two time windows:

$$r_1 = \theta_{10} + \theta_{1A}\, q_A\, \delta_{order,AB} + \theta_{1B}\, q_B\, \delta_{order,BA} \tag{11}$$

$$r_2 = \theta_{20} + \theta_{2A}\, q_A\, \delta_{order,BA} + \theta_{2B}\, q_B\, \delta_{order,AB} \tag{12}$$

where $r_1$ and $r_2$ were their responses recorded in the post-offer1 and post-offer2 time windows, respectively, and $\theta_{10}$, $\theta_{1A}$, $\theta_{1B}$, $\theta_{20}$, $\theta_{2A}$, and $\theta_{2B}$ were regression coefficients. These coefficients provided four neuronal measures of relative value:

$$\rho^{neuronal}{}_{offer1} = \theta_{1A}/\theta_{1B} \tag{13}$$

$$\rho^{neuronal}{}_{offer2} = \theta_{2A}/\theta_{2B} \tag{14}$$

$$\rho^{neuronal}{}_{AB} = \theta_{1A}/\theta_{2B} \tag{15}$$

$$\rho^{neuronal}{}_{BA} = \theta_{2A}/\theta_{1B} \tag{16}$$

In essence, these four measures corresponded to the two time windows (post-offer1 and post-offer2) and to the two presentation orders (AB and BA). Importantly, all these measures were computed conditioned on $\theta_{1A}$, $\theta_{1B}$, $\theta_{2A}$, and $\theta_{2B}$ differing significantly from zero ($p < 0.05$). The analyses illustrated in ***Figures 5 and 7*** were restricted to neurons satisfying this criterion.

In terms of notation, we often omit the superscript in $\rho^{behavioral}$ and we indicate behavioral measures simply as $\rho$ (with the relevant subscripts). We use the superscript 'behavioral' only when we explicitly compare behavioral and neuronal measures, for clarity. In contrast, for neuronal measures of relative value we always use the superscript 'neuronal'.

## Activity profiles of chosen juice cells

To conduct population analyses, we pooled all *chosen juice* cells. The juice eliciting higher firing rates was labeled 'E' (encoded) and other juice was labeled 'O'. In Task 2, we thus referred to EO trials and OE trials, depending on the presentation order.

To illustrate the activity profiles of *chosen juice* cells in Task 2, we aligned spike trains at offer1 and, separately, at juice delivery. For each trial, the spike train was smoothed using a kernel that mimicked the postsynaptic potential by exerting influence only forward in time (decay time constant = 20 ms) (*So and Stuphorn, 2010*). In *Figures 6 and 8A*, we used moving averages of 100 ms with 25 ms steps for display purposes.

Under sequential offers, *chosen juice* cells encode different variables in different time windows (see *Table 1*). During offer1 and offer2 presentation, these cells encode in a binary way the juice type currently on display. Later, as the decision develops, these neurons gradually come to encode the binary choice outcome (i.e., the chosen juice). We previously showed that the activity of these neurons recorded in OE trials shortly before offer2 is inversely related to the value of offer1 (*Ballesta and Padoa-Schioppa, 2019*). This phenomenon, termed circuit inhibition, resembles the setting of a dynamic system's initial conditions and is regarded as an integral part of the decision process (*Ballesta and Padoa-Schioppa, 2019*).

For a quantitative analysis of circuit inhibition, we focused on a 300-ms time window starting 250 ms before offer2 onset. We excluded forced choice trials, for which one of the two offers was null. For each neuron, we examined OE trials and we regressed the firing rates against the normalized value of offer1:

$$r = c_0 + c_1 \, V(O)/\Delta V_o \tag{17}$$

where $\Delta V_O$ was the value range for juice O. The normalization allowed to pool neurons recorded with different value ranges. The regression slope $c_1$ quantified circuit inhibition for individual cells, and we studied this parameter at the population level.

The activity of *chosen juice* cells in OE trials captures the momentary state of the decision and thus the evolving commitment to a particular choice outcome. To quantify the momentary decision state, we conducted an ROC analysis (*Green and Swets, 1966*) on the activity recorded during OE trials. This analysis was conducted on raw spike counts, without kernel smoothing, time averaging or baseline correction. We restricted the analysis to offer types for which the animal split choices between the two juices and we excluded trial types with <2 trials. For each offer type, we divided trials depending on the chosen juice (E or O) and we compared the two distributions. The ROC analysis provided an area under the curve (AUC). For each neuron, we averaged the AUC across offer types to obtain the overall CP (*Kang and Maunsell, 2012*). The ROC analysis was performed in 100ms time windows shifted by 25 ms. We also conducted the same analysis on four 250 ms time windows, namely pre-offer1 (−250 to 0 ms from offer1 onset), late offer2 (−250 to 0 ms from offer1 offset), early wait (0 to 250 ms after offer2 offset), and pre-juice (−250 to 0 ms before juice delivery) (*Figure 8*). In *Figures 6 and 8B–I*, cells were excluded because the Matlab function *perfcurve.m* failed to converge.

## Acknowledgements

We thank H Schoknecht for help with animal training, L Snyder for helpful discussions, and Z Balewski, E Bromberg-Martin, K Conen, A Livi, P Natenzon, T Ott, J Tu, and M Zhang for comments on the manuscript. This research was supported by the National Institutes of Health (grant number R01-MH104494 to CPS) and by the McDonnell Center for Systems Neuroscience (predoctoral fellowship to WS).

## Additional information

### Funding

| Funder | Grant reference number | Author |
|---|---|---|
| National Institute of Mental Health | R01-MH104494 | Camillo Padoa-Schioppa |
| McDonnell Center for Systems Neuroscience | CCSN Fellowship | Weikang Shi |

The funders had no role in study design, data collection, and interpretation, or the decision to submit the work for publication.

### Author contributions

Weikang Shi, Conceptualization, Data curation, Formal analysis, Investigation, Software, Visualization, Writing – original draft, Writing – review and editing; Sebastien Ballesta, Conceptualization, Data curation, Writing – review and editing; Camillo Padoa-Schioppa, Conceptualization, Formal analysis, Funding acquisition, Project administration, Supervision, Writing – original draft, Writing – review and editing

### Author ORCIDs

Weikang Shi ⬥ http://orcid.org/0000-0002-4068-1168
Camillo Padoa-Schioppa ⬥ http://orcid.org/0000-0002-7519-8790

### Ethics

All the experimental procedures adhered to the NIH Guide for the Care and Use of Laboratory Animals and were approved by the Institutional Animal Care and Use Committee (IACUC) at Washington University (protocol number 190931).

### Decision letter and Author response

Decision letter https://doi.org/10.7554/eLife.75910.sa1
Author response https://doi.org/10.7554/eLife.75910.sa2

## Additional files

### Supplementary files
• Transparent reporting form

### Data availability

Neuronal data and analysis scripts are deposited in GitHub: https://github.com/PadoaSchioppaLab/2022_eLife_choicebias, (copy archived at swh:1:rev:a474955590576fecabc02fac14edfc2ef4e89144).

The following dataset was generated:

| Author(s) | Year | Dataset title | Dataset URL | Database and Identifier |
|---|---|---|---|---|
| Padoa-Schioppa C | 2022 | Neuronal origins of reduced accuracy and biases in economic choices under sequential offers | https://github.com/PadoaSchioppaLab/2022_eLife_choicebias | GitHub, PadoaSchioppaLab |

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
