## [Editor Report]

This manuscript describes three decision biases in value-based choice paradigms. Building on previous work from the lab, the authors focus on neural coding of decision variables in the orbitofrontal cortex of rhesus monkeys, and convincingly argue that different biases arise at different stages of the decision-making process. The reviewers found the study rigorous and believe that the results will be of broad interest. Understanding the neural mechanisms that produce biases in decision-making is an important goal for the field of decision-making and neuroeconomics, and also has relevance to conditions that involve disordered decision-making.

---

## [Decision Letter]

**Decision letter after peer review:**

Thank you for submitting your article "Neuronal origins of reduced accuracy and biases in economic choices under sequential offers" for consideration by *eLife*. Your article has been reviewed by 3 peer reviewers, and the evaluation has been overseen by a Reviewing Editor and Michael Frank as the Senior Editor. The following individual involved in review of your submission has agreed to reveal their identity: Veit Stuphorn (Reviewer #3).

Overall, reviewers found the study convincing and the results a significant advance in understanding circuit-level computations underlying decision-making. However, they raised multiple important issues to improve the clarity and specificity of the claims. The most important points to be addressed in revision are listed below, and the reviewers' individual comments are attached in full to provide additional context and suggestions.

Essential revisions:

1. The main topic that arose in reviewer consultation was the degree to which the conclusions rely on the lab's previous work parsing neural responses into discrete categories. Whereas this work has shown the reliability of these categories within the lab's own data, other literature has found conflicting results. Given that the degree to which categorical boundaries can be set within a continuum of response profiles is debatable, the reviewers think it's important to include discussion of the issue and acknowledgement of the explicit assumptions of the authors' methods and framework. Importantly, it's not just the categorization that the interpretation relies on: it's the assumption that each plays specific decision roles (valuation vs. comparison). (R1/R2/R3)

2. Conceptualizing the two tasks as computationally identical may not be accurate. The authors should discuss potential roles of working memory in task 2, and assess their model fits to behavior in each task independently. (R2/R3)

3. The idea of optimality, bias, and what constitutes reduced accuracy was also raised, and should be addressed by clarifications of the text: e.g. describe how exactly the different choice effects are 'detrimental', whether effects should be considered biases versus stochasticity, and how the findings here relate to classic effects in economic choice. (R2)

4. Please describe the rationale behind the order bias calculation, as opposed to using the a4 coefficient. In addition, consider how the signs of the neural order biases predict a consistent behavioral order bias. (R1/R2)

5. Please address why output signals do not reflect the reduced value sensitivity of the input signals. (R3)

*Reviewer #1 (Recommendations for the authors):*

1. The core premise of their interpretation/model is that the functional cell types they define (offer value, chosen value cells) are (mostly?) non-overlapping populations, if they contribute to distinct aspects of behavior. Is that strictly true, or are there subsets of cells that belong to both groups? In the methods section, it sounds like they force neurons to belong to only one group ("Tuned cells were assigned to the cell class that provided the maximum |sum(sR2 )final|"). This decision seems arbitrary but central to the interpretation that these really are different functional cell types.

2. On page 7, they define the order bias as ε = 2 ρTask2 a4/a3. Why don't they just evaluate the magnitude of the a4 coefficient? If it were 0, wouldn't that indicate no order bias? The sign would indicate the bias for first vs. second, and the magnitude the strength of the bias? Perhaps I am missing something, but that seems like a much more straightforward way to obtain an estimate of order bias from this model.

3. The authors show that offer value cells exhibit reduced responses for sequential compared to simultaneous offers. Is this true for both offer1 and offer2? They show an example neuron for both offer1 and offer2, but the population activity pools those responses together.

4. Related to point #1, it sounds like they combine data from task 1 and task 2 when they determine whether a cell is an "offer cell" or a "chosen cell." Are most of the offer value cells categorized based on their task 1 responses (i.e., is the maximum |sum(sR2 )final| the task1 offer value)? I am wondering the following: because there is a pre-stimulus cue at the beginning of each trial that tells the monkey which task it is performing, it's possible that the monkey/OFC treats them as two different tasks, and that slightly different populations of neurons encode offer value in each case. If that were the case, then using the task 1 responses to define the cells as encoding offer value could trivially result in the first finding, that there are reduced offer value responses in task 2. What if the authors determined whether cells were offer value cells based on the task 2 responses only? Does the result hold that the offer value responses are reduced, relative to task 1?

5. Figure 4F, the correlation is not very compelling. Is there something different about the sessions in which the sigmoid steepness and δ activity range go in opposite directions (i.e., the upper left and lower right quadrants?)

6. Throughout the paper, it is hard to remember what the different Greek letters refer to. For instance, on page 13, when ε reappears, I couldn't remember what that referred to. When possible, for clarity it might help to use interpretable terms when possible (e.g., order bias instead of ε).

*Reviewer #2 (Recommendations for the authors):*

Much of my feedback can likely be gleaned from the public review, but I'll be specific (and probably repetitive) here:

(1) It would be helpful if the authors provided a more detailed framing of existing behavioral economic choice biases and how the current choice effects relate. For terminology, it is a bit confusing that these previous findings are labelled "biases" but that one of the effects discussed here is increased stochasticity (reduced accuracy) rather than a bias – can the authors find some way of being more general about choice effects?

(2) For the analysis of choice behavior and the use of probit regression, it would help if the authors could quantify the goodness of fit of their choice functions (probit model using log amount ratios), specifically against alternative models (logit, and/or models with amount difference rather than log ratio). The big issue is that the relevant measures (accuracy, order bias, preference bias) depend largely on parameters derived form these fits, so it is important to see that the fits good and the model is appropriate. for the order effect in particular, have the authors examined the data while relaxing the assumption of a constant sigmoid?

(3) Re: the documentation of the preference bias, I would like the authors to be clearer in the text about the statistical test of the basic preference effect (higher rho in Task 1 vs. Task 2) – the effect is rather strong, and just needs to be clearly conveyed in the text. I do think that the dependence of the effect on relative value (i.e. rotation of the ellipse) seems to be less well supported: beyond clearing up the presentation of the significance of the results, perhaps this could be places in a clearer context vis a vis the basic preference bias – I don't think it is necessary to the general findings.

(4) Please be clearer about how exactly the different effects are detrimental. I think this point related directly to the initial framing about the difficulty of characterizing the optimality of value-based choice using behavior alone.

(5) Regarding the interpretation of OFC activity within the Rustichini/Wang modeling framework, I realize that the authors have good reason (and past data) to adopt this interpretation – I think it would be fair to at least acknowledge the explicit assumptions for the general reader.

(6) One interpretation issue that remains is whether the OFC neural effects are causal or correlational (particularly the accuracy and order effects, which can be seen in OFC activity). One issue that points towards the latter, at least in the order bias, is the sign of the neuronal effects (Figure 5C, Figure 6C). While the behavioral order bias is overwhelming positive, the neural effects (difference in AB vs BA rhos, circuit inhibition) do not on average differ from zero, even if they are correlated with the behavioral effect. If these neural effects drive the bias, shouldn't hey also be biased in the correct direction? Can the authors provide an interpretation?

(7) Additional points:

– How exactly is the preferred item (A vs. B) defined – across both tasks combined?

– Given the initial framing, it seems to me that these data – especially the preference bias effect – provides insight into the goods versus action based choice discussion (and relates to some previous work like Cai and Padia-Schioppa). Perhaps it would be relevant to discuss this briefly at the end of the paper.

– A conceptual question: given that all three effects are presented as suboptimal, why should this be the case given the initial framing that sequential decisions are the prevalent, natural form of choices?

*Reviewer #3 (Recommendations for the authors):*

The paper is written very clearly and the underlying logic for the different analysis is very well described. The results are for the most part convincing, albeit sometimes of a weak effect size.

---

## [Author Response]

Essential revisions:1. The main topic that arose in reviewer consultation was the degree to which the conclusions rely on the lab's previous work parsing neural responses into discrete categories. Whereas this work has shown the reliability of these categories within the lab's own data, other literature has found conflicting results. Given that the degree to which categorical boundaries can be set within a continuum of response profiles is debatable, the reviewers think it's important to include discussion of the issue and acknowledgement of the explicit assumptions of the authors' methods and framework. Importantly, it's not just the categorization that the interpretation relies on: it's the assumption that each plays specific decision roles (valuation vs. comparison). (R1/R2/R3)

The reviewers are correct. Our analyses were designed, and the results were interpreted, under a series of assumptions:

(a) Good-based decisions take place in OFC

(b) Different groups of neurons in OFC encode categorically distinct variables

(c) Choices under simultaneous and sequential offers engage the same cell groups

(d) These cell groups constitute the building blocks of a decision circuit. Specifically, offer value cells provide input to a circuit composed of chosen value cells and chosen juice cells, where values are compared and the decision is formed (Figure 1)

Assumption (a) builds on a broad literature and is generally accepted in the field (Cisek, 2012; Padoa-Schioppa, 2011; Rushworth et al., 2012). It is also supported by recent results showing that weak electrical stimulation of OFC selectively disrupts value comparison without inducing any choice bias (Ballesta et al., 2021). Assumptions (b) and (c) are directly supported by a series of studies showing categorical encoding in OFC (Hirokawa et al., 2019; Onken et al., 2019; Padoa-Schioppa, 2013) and by previous analysis of the same data examined here (Shi et al., 2022). In contrast, assumption (d) remains a working hypothesis, in the sense that the organization and mechanisms of the decision circuit are not well understood. Importantly, the scheme proposed here is very general in the sense that we don’t make any particular assumption about the connections between the cell groups, except that *offer value* cells are upstream of the other cell groups. This point is supported by analyses correlating choice variability with trial-by-trial variability in the activity of *offer value* cells (Conen and Padoa-Schioppa, 2015). However, we recognize that more work is needed to ascertain the anatomical organization of the decision circuit.

We revised the manuscript including in the Discussion the following paragraph that summarizes these points:

“In our experiments, monkeys chose between two juices offered simultaneously or sequentially. Choices under sequential offers were less accurate, biased in favor of the second offer (order bias), and biased in favor of the preferred juice (preference bias). It is generally understood that good-based economic decisions take place in OFC (Cisek, 2012; Padoa-Schioppa, 2011; Rushworth et al., 2012) and that the encoding of decision variables in this area is categorical in nature (Hirokawa et al., 2019; Onken et al., 2019; Padoa-Schioppa, 2013). Earlier studies had identified in OFC three groups of neurons encoding individual offer values, the chosen juice and the chosen value. Importantly, choices under simultaneous or sequential offers engage the same neurons (Shi et al., 2022). Notably, the variables encoded in OFC capture both the input and the output of the decision process. This observation and a series of experimental (Ballesta et al., 2021; Rich and Wallis, 2016) and theoretical results lead to the hypothesis that the cell groups identified in OFC constitute the building blocks of a decision circuit (Padoa-Schioppa and Conen, 2017). In this view, *offer value* cells provide the primary input to a circuit formed by *chosen juice* cells and *chosen value* cells, where decisions are formed. Different cell groups in OFC may thus be associated with different computational stages: *offer value* cells instantiate the valuation stage; *chosen value* cells reflect values possibly modified by the decision process; and *chosen juice* cells capture the evolving commitment to a particular choice outcome. In this framework, we examined the activity of each cell group in relation to each behavioral phenomenon.

[…] As a caveat, our results rely on the hypothesis that different cell groups identified in OFC play specific roles in the decision process. This working hypothesis awaits further confirmation.”

2. Conceptualizing the two tasks as computationally identical may not be accurate. The authors should discuss potential roles of working memory in task 2, and assess their model fits to behavior in each task independently. (R2/R3)

Actually, we don’t describe the two tasks as identical. In fact, we discuss how Task 2 involves a series of cognitive operations not required in Task 1, each of which could in principle introduce noise or biases (p.6). That said, we do argue that the two tasks engage the same groups of neurons in OFC. This statement is based on empirical evidence – that is, previous analyses of this same data set (Shi et al., 2022). We elaborate on this issue below in response to R1 (Major comments, point 1).

3. The idea of optimality, bias, and what constitutes reduced accuracy was also raised, and should be addressed by clarifications of the text: e.g. describe how exactly the different choice effects are 'detrimental', whether effects should be considered biases versus stochasticity, and how the findings here relate to classic effects in economic choice. (R2)

We added to the Discussion a new section ‘*The cost of choice biases*’ (p.13) that elaborates on this important issue. In a nutshell, if in two conditions (e.g., Task1 and Task2) subjective values are the same but choices are different, in one or both conditions the subject fails to choose the higher value. In that sense, the choice bias is detrimental. Our analyses of neuronal activity indicated that subjective offer values were the same in the two tasks, and were the same independent of the offer presentation in Task 2. Hence, both the preference bias and the order bias were detrimental to the animal.

4. Please describe the rationale behind the order bias calculation, as opposed to using the a4 coefficient. In addition, consider how the signs of the neural order biases predict a consistent behavioral order bias. (R1/R2)

We added to the Methods a new section ‘*Behavioral analyses*’ (p.16-17) where we discuss alternative logistic analyses (value difference instead of log value ratio; logit instead of probit). In the same section, we also explain the rationale for defining the order bias as we did, and we report that control analyses based on alternative definitions of the order bias provided essentially the same results.

5. Please address why output signals do not reflect the reduced value sensitivity of the input signals. (R3)

We agree that this result was puzzling and we gave this issue more thought. We realized that our definition of activity range for chosen value cells was arbitrary and – in some sense – not consistent across tasks. In Task 2, in the time windows of interest for this analysis (post-offer1 and post-offer2), these neurons encode the value of the offer currently on display. That value ranges between zero (forced choices for the other good) and the max offer value. In contrast, in Task 1, in the relevant time window (post-offer), these neurons encode the chosen value. That value ranges between the minimum chosen value (which is always >0) and the max offer value. In our previous definition of activity range, we had overlooked this difference. We have now corrected our definition, defining the activity range as that corresponding to the range of values [0, max offer value] in both tasks. According to this new and better definition, the activity range of *chosen value* cells is reduced in Task 2 compared to Task 1. Of note, this issue is not relevant to *offer value* cells, because for those neurons the minimum encoded value is always =0 in both tasks. The revised manuscript reports the new results (Figure 4—figure supplement 1).

Reviewer #1 (Recommendations for the authors):1. The core premise of their interpretation/model is that the functional cell types they define (offer value, chosen value cells) are (mostly?) non-overlapping populations, if they contribute to distinct aspects of behavior. Is that strictly true, or are there subsets of cells that belong to both groups? In the methods section, it sounds like they force neurons to belong to only one group ("Tuned cells were assigned to the cell class that provided the maximum |sum(sR2 )final|"). This decision seems arbitrary but central to the interpretation that these really are different functional cell types.

This issue is partly addressed above, under Essential Revisions, point (1). Here we would like to emphasize a few points.

First, this study relies on previous work in monkeys (Onken et al., 2019; Padoa-Schioppa, 2013) and rodents (Hirokawa et al., 2019) showing that the encoding of decision variables in OFC is categorical in nature. In other words, in a statistical sense, different groups of neurons encode different variables. Of course, our ability to assess the class of a particular cell is limited by neuronal noise, a finite number of trials, and the fact that the encoded variables are correlated with each other. Hence, we certainly make some classification errors. In previous work (Xie and Padoa-Schioppa, 2016), we estimated our classification precision to be ~80%.

Second, previous analyses of this same data set indicated that the cell groups identified in the two tasks are one and the same (Shi et al., 2022).

Third, it is clear that classifying cells using trials from both tasks will (a) reduce our classification errors and (b) avoid biasing any further analysis in favor of one task of the other. Indeed, this is why we classified neurons using both tasks.

Finally, while it is true that the idea of distinct cell groups is central to our analyses and thus to the interpretation, classification errors are not a confounding factor for any of our results. Indeed, classification errors are like noise, and will generally make it harder – not easier – to demonstrate differences between cell groups and correlations between behavioral measures and neuronal measures derived for a particular cell group.

These considerations leave us confident that our results are valid and not artifactual.

2. On page 7, they define the order bias as ε = 2 ρTask2 a4/a3. Why don't they just evaluate the magnitude of the a4 coefficient? If it were 0, wouldn't that indicate no order bias? The sign would indicate the bias for first vs. second, and the magnitude the strength of the bias? Perhaps I am missing something, but that seems like a much more straightforward way to obtain an estimate of order bias from this model.

We added to the Methods a new section ‘Behavioral analyses’ (p.16-17). There we clarified that the definition of order bias used here (*ε = 2 ρ_Task2_ a_4_/a_3_*) is particularly convenient for the present analyses because *ε* equals the difference *ρ_BA_ – ρ_AB_* (Equation 3). Alternative and valid definitions include *ε=a_4_* and *ε=a_4_/a_3_*. To address this question from R1, we conducted a series of control analyses using these definitions, and found that the results reported in the manuscript did not vary in any substantial way (not shown).

3. The authors show that offer value cells exhibit reduced responses for sequential compared to simultaneous offers. Is this true for both offer1 and offer2? They show an example neuron for both offer1 and offer2, but the population activity pools those responses together.

The short answer is that the result held for each time window, but it was statistically significant only for post-offer1. Author response image 1 illustrates this point. For a control, we tested whether the drop in activity range measured in Task 2 depended on the time window. Thus we computed the differences in activity range ΔAR_post-offer1_ = AR_Task 2, post-offer1_ – AR_Task 1, post-offer_ and ΔAR_post-offer2_ = AR_Task 2, post-offer2_ – AR_Task 1, post-offer_ and we examined the relationship between these measures. We did not find any significant difference (Author response image 1) .

**Author response image 1. sa2fig1:** Weaker offer value signals in Task 2, population analysis in individual time windows. (AB) Post-offer1 time window (N = 53 *offer value* cells). (CD) Post-offer2 time window (N = 56 offer value cells). Panel A-D are in the same format as Figure 4EF. (E) Comparing the effect across time windows. X- and y-axis represent ΔAR_post-offer1_ and ΔAR_post-offer2_, respectively. Across the population, the two measures were statistically indistinguishable.

4. Related to point #1, it sounds like they combine data from task 1 and task 2 when they determine whether a cell is an "offer cell" or a "chosen cell." Are most of the offer value cells categorized based on their task 1 responses (i.e., is the maximum |sum(sR2 )final| the task1 offer value)? I am wondering the following: because there is a pre-stimulus cue at the beginning of each trial that tells the monkey which task it is performing, it's possible that the monkey/OFC treats them as two different tasks, and that slightly different populations of neurons encode offer value in each case. If that were the case, then using the task 1 responses to define the cells as encoding offer value could trivially result in the first finding, that there are reduced offer value responses in task 2. What if the authors determined whether cells were offer value cells based on the task 2 responses only? Does the result hold that the offer value responses are reduced, relative to task 1?

This issue is fully addressed above (point 1). The key point is that we classify cells using information from both tasks – i.e., in an unbiased way. We think that this resolves the issue. That said, we repeated the analysis of Figure 4 having classified neurons only on the basis of Task 2. Although the correlation was understandably weaker, the result of Figure 4 held true (Author response image 2) .

**Author response image 2. sa2fig2:** Weaker offer value signals in Task 2, population analysis based on a neuronal classification relying only on Task 2 (N = 74 offer value cells). Panel A and B are in the same format as Figure 4EF.

5. Figure 4F, the correlation is not very compelling. Is there something different about the sessions in which the sigmoid steepness and δ activity range go in opposite directions (i.e., the upper left and lower right quadrants?)

As discussed in the two previous points, the result shown in Figure 4F is actually quite robust. Of course, those examined here are noisy neuronal measures. We are not aware of anything that would differentiate sessions or cells populating the “wrong” quadrants (2^nd^ and 4^th^) compared to the “right ” ones (1^st^ and 3^rd^).

6. Throughout the paper, it is hard to remember what the different Greek letters refer to. For instance, on page 13, when ε reappears, I couldn't remember what that referred to. When possible, for clarity it might help to use interpretable terms when possible (e.g., order bias instead of ε).

In the revised manuscript, we added several reminders about what Greek letters stand for.

Reviewer #2 (Recommendations for the authors):Much of my feedback can likely be gleaned from the public review, but I'll be specific (and probably repetitive) here:(1) It would be helpful if the authors provided a more detailed framing of existing behavioral economic choice biases and how the current choice effects relate. For terminology, it is a bit confusing that these previous findings are labelled "biases" but that one of the effects discussed here is increased stochasticity (reduced accuracy) rather than a bias – can the authors find some way of being more general about choice effects?

See Public review, weaknesses, point 1.

(2) For the analysis of choice behavior and the use of probit regression, it would help if the authors could quantify the goodness of fit of their choice functions (probit model using log amount ratios), specifically against alternative models (logit, and/or models with amount difference rather than log ratio). The big issue is that the relevant measures (accuracy, order bias, preference bias) depend largely on parameters derived form these fits, so it is important to see that the fits good and the model is appropriate. for the order effect in particular, have the authors examined the data while relaxing the assumption of a constant sigmoid?

Thanks for raising this question. To address it, we conducted a series of control analyses.

First, we kept the assumption of parallel sigmoids and we repeated the behavioral analysis using different regression functions (logit instead of probit) and different models (value difference instead of log value ratio). For each fit, we derived measures of the relative value (in each task), the sigmoid steepness (in each task) and the order bias (in Task 2), as well as a measure for the goodness of fit. Importantly, the measures obtained for different fits were all very similar. To compare the goodness of fit obtained with different functions and models, we used the Bayesian information criterion (BIC). Author response image 3 illustrates the 3 most relevant comparisons (pooling data for Task 1 and Task 2). In essence, the logit typically provided a better fit than the probit, and the log value ratio model typically provided a better fit than the value difference model. The latter result is consistent with theoretical considerations (Padoa Schioppa, 2022).

**Author response image 3. sa2fig3:** Comparing behavioral models (N = 241 sessions, pooled Task 1 and Task 2). (A) BIC difference between probit regression with log ratio vs.. logit regression with log ratio. (B) BIC difference between probit regression with log ratio vs.. probit regression with linear difference. (C) BIC difference between logit regression with log ratio vs.. logit regression with linear difference. Smaller BIC indicates better fitting goodness, therefore negative BIC difference indicates the former model was better and vis versa. Altogether, logit regression with log quantity ratio shows the best goodness of fit. Individual data from Task 1 or Task 2 alone show similar results (not shown).

Second, we repeated all the analyses relating neuronal activity to behavioral measures using measures derived from the logit function using the log value ratio model. We found that all the results held true.

Third, we examined whether assuming parallel sigmoids for AB and BA trials in Task 2 affected the results. Thus we fitted the two groups of trials with independent sigmoids. As illustrated in Author response image 4, we did not find any systematic difference between the two measures of steepness. We went on and repeated the analyses of neuronal activity using behavioral measures obtained fitting independent sigmoids. None of the results presented in the manuscript were substantially affected.

**Author response image 4. sa2fig4:** No steepness difference between AB and BA trials in Task 2 (N = 241 sessions, pooled two monkeys).

The revised manuscript includes a new section in the Methods (‘Behavioral analysis’) where we discuss these issues and describe the results of control analyses.

(3) Re: the documentation of the preference bias, I would like the authors to be clearer in the text about the statistical test of the basic preference effect (higher rho in Task 1 vs. Task 2) – the effect is rather strong, and just needs to be clearly conveyed in the text. I do think that the dependence of the effect on relative value (i.e. rotation of the ellipse) seems to be less well supported: beyond clearing up the presentation of the significance of the results, perhaps this could be places in a clearer context vis a vis the basic preference bias – I don't think it is necessary to the general findings.

See Public review, weaknesses, point 3.

(4) Please be clearer about how exactly the different effects are detrimental. I think this point related directly to the initial framing about the difficulty of characterizing the optimality of value-based choice using behavior alone.

See Public review, weaknesses, point 4.

(5) Regarding the interpretation of OFC activity within the Rustichini/Wang modeling framework, I realize that the authors have good reason (and past data) to adopt this interpretation – I think it would be fair to at least acknowledge the explicit assumptions for the general reader.

See Public review, weaknesses, point 5.

(6) One interpretation issue that remains is whether the OFC neural effects are causal or correlational (particularly the accuracy and order effects, which can be seen in OFC activity). One issue that points towards the latter, at least in the order bias, is the sign of the neuronal effects (Figure 5C, Figure 6C). While the behavioral order bias is overwhelming positive, the neural effects (difference in AB vs BA rhos, circuit inhibition) do not on average differ from zero, even if they are correlated with the behavioral effect. If these neural effects drive the bias, shouldn't hey also be biased in the correct direction? Can the authors provide an interpretation?

We agree that this is an important issue. Aside from other considerations, the experiments and analyses conducted in this study can only show correlations – not causal relations – between neural activity and behavioral effects. Having recently conducted causal experiments, we are very primed to this distinction, and we think that our writing throughout this manuscript is unambiguous about this issue. That said, the signs in Figure 5C and Figure 6C brought up by R2 deserve some discussion.

With respect to Figure 6C, the location of the data cloud on the axis is actually consistent with our predictions and understanding. The reason is that we expect some amount of circuit inhibition (y-axis) even in the absence of order bias. In fact, we think that this amount of circuit inhibition is critical to the decision process under sequential offers (Ballesta and Padoa-Schioppa, 2019). In this view, reduced circuit inhibition would correlate with the order bias. Consistently, the regression line that summarizes the data cloud has a negative intercept (y<0 for x=0; presence of circuit inhibition when the order bias = 0) and positive slope (reduced circuit inhibition correlating with positive order bias).

In contrast, the cloud of data points in Figure 5C differs somewhat from our natural intuition. Specifically, the regression line has a negative intercept (y<0 for x=0). In other words, the entire data cloud is displaced downwards compared to where one might have thought. As a result, if we project the whole cloud on the y axis, the center of the distribution does not differ significantly from zero. As we conclude this study, we don’t have a good interpretation for this phenomenon, and future work will need to revisit this issue. We clarified this point in the figure legend.

(7) Additional points:– How exactly is the preferred item (A vs. B) defined – across both tasks combined?

Juice A was defined such that (ρ_Task1_ + ρ_Task2_)/2 ≥ 1. (We typically knew ahead of time what juice was preferred; in rare cases where the inequality turned out in the opposite direction, we inverted the definition of A and B.)

– Given the initial framing, it seems to me that these data – especially the preference bias effect – provides insight into the goods versus action based choice discussion (and relates to some previous work like Cai and Padia-Schioppa). Perhaps it would be relevant to discuss this briefly at the end of the paper.

The debate about action-based vs good-based decisions has traditionally been about whether value comparison takes place in the space of actions of in the space of goods. The present results do not revisit that issue – in fact, we start from the assumption that values are compared in goods space. That said, even if one embraces the good-based model, as we do, it is clear that the choice outcome has to be transformed into a suitable action. To this point, which we made many other times, the present study adds a simple consideration: noise and biases affecting choices could also emerge late, after value comparison. Furthermore, we show that the preference bias indeed emerged late. This result might be perceived as resonating with the idea of a “distributed consensus” involving multiple representations (Cisek, 2012). However, in our understanding of the distributed consensus model, different representation would be engaged in parallel and not serially as we argue here.

– A conceptual question: given that all three effects are presented as suboptimal, why should this be the case given the initial framing that sequential decisions are the prevalent, natural form of choices?

Great question, and we don’t have a definite answer. But here are a few thoughts. In the behavioral economics literature, there has long been the idea that acquiring information or paying attention to all aspects of the offers is costly, and thus in some situations it may be rationale to make choices that appear affected by systematic errors (i.e., biases) but that reduce the attentional cost (Simon, 1956; Sims, 2003). Along similar lines, the two biases documented here might reflect some ecological trade-off. For example, maintaining in working memory the value of offer1 until offer2 must have some metabolic cost. The animal would be better off remembering offer1 faultlessly and choosing the higher value, but there is some trade-off between the metabolic cost of working memory and the cost of choosing the lower value once in a while. And if the animal forgets the value of offer1, it makes sense to bias the choice in favor of offer2 (order bias). The important point is that this trade-off involving metabolic costs does not enter explicitly the encoding and the comparison of offer values. In other words, insofar as the metabolic cost of remembering offer1 affects choices, it does so in a meta-decision sense. Similar arguments can be made apropos the preference bias and the drop in choice accuracy. In the revised Discussion, we included a new section ‘*The cost of choice biases*’, where we discuss these issues.